# Population dynamics of modern planktonic foraminifera in the western Barents Sea

Julie Meilland[1], Hélène Howa[2], Vivien Hulot[3,4], Isaline Demangel[5,6], Joëlle Salaün[7], Thierry Garlan[7]

[1] MARUM - Center for Marine Environmental Sciences, Leobener Str. 8, D-28359, Bremen, Germany

[2] LPG-BIAF, UMR-CNRS 6112; University of Angers, France

[3] University of French Polynesia, UMR-241 EIO, Labex Corail, FAA'A, Tahiti, French Polynesia

[4] Ifremer, UMR-241 EIO, Labex Corail, Departement Ressources Biologiques et Environnement, Vairao, Tahiti, French Polynesia

[5] Institute of Earth Sciences, University of Graz, NAWI Graz Geocenter, Graz, Austria

[6] Department of Geology, University of Lund, Lund, Sweden

[7] SHOM –Sciences et Techniques Marines/Géologie Marine, Brest, France

*Correspondence to*: Julie Meilland (jmeilland@marum.de)

**Abstract.** This study reports on diversity and distribution of planktonic foraminifera (PF) in the Barents Sea Opening (BSO). Populations of PF living in late summer (collected by means of stratified plankton tows) and recently deposited individuals (sampled by interface corer) were compared. High abundances reaching up to 400 ind.m$^{-3}$ in tow samples and 8000 ind.cm$^{-3}$ in surface sediments were recorded in the centre of the studied area while low abundances were observed in coastal areas, likely due to continental influences. The living and core-top assemblages are mainly composed of the same four species *Neogloboquadrina pachyderma, Neogloboquadrina incompta*, *Turborotalita quinqueloba* and *Globigerinita uvula*. The two species *G. uvula* and *T. quinqueloba* dominate the upper water column, whereas surface sediment assemblages display particularly high concentrations of *N. pachyderma*. The unusual dominance of *G. uvula* in the water sample assemblages compared to its low proportion in surface sediments might be the signature of 1) a seasonal signal due to summer phytoplankton composition changes at the BSO, linked to the increase of summer temperature at the study site, and/or 2) a signal of a larger time-scale and wider geographical reach phenomenon reflecting poleward temperate/subpolar species migration and consecutive foraminiferal assemblage diversification at high latitudes due to global change. Protein concentrations were measured on single specimens and used as a proxy of individual carbon biomass. Specimens of all species show the same trend, a northward decrease of their size-normalized-protein concentration. This suggests that foraminiferal biomass is potentially controlled by different constituents of their organelles (e.g. lipids). The coupling of data from plankton tows, protein measurements and surface sediments allows us to hypothesise that PF dynamics (seasonality and distribution) are decoupled from their metabolism.

**Keywords:** planktonic foraminifera, living and dead communities, latitudinal distribution, protein content, seasonality, atlantification, Barents Sea

## 1 Introduction

Polar areas are sensitive to global temperature changes, particularly in the Arctic where warming occurs faster than in the rest of the world and has accelerated over the past 50 years (Shepherd, 2016). This Arctic amplification appears to be mainly caused by sea-ice loss under increasing $CO_2$ (Dai et al., 2019). Recently increased advection of Atlantic Water into the Barents Sea modifies its physico-chemical properties (Smedsrud et al., 2013), which directly affect the entire ecosystems in the region. Higher temperatures lead to increased rates of planktonic primary production (Vaquer-Sunyer et al., 2013) and increased $CO_2$ concentrations are expected to have a fertilization effect on marine autotrophs (Holding et al., 2015). Not only is productivity increasing, but spring and summer blooms are also occurring earlier in the European Arctic Ocean (Oziel et al., 2017). As a response, some taxa of calcifying groups (i.e. foraminifera, coccolithophores, molluscs and echinoderms; Beaugrand et al., 2013) exhibit a poleward movement in agreement with expected biogeographical changes under sea temperature warming. Both satellite images (Smyth et al., 2004; Burenkov et al., 2011) and *in situ* measurements (Dylmer et al., 2013; Giraudeau et al., 2016; AMAP 2018) have shown rapid expansion of temperate species of coccolithophores in the Arctic. For example, *Emiliania huxleyi* shows a striking poleward shift (>5°) in the distribution of its blooms (Neukermans et al., 2018). This phenomenon, called "atlantification" (Årthun et al., 2012), is expected to impact every trophic level of the food web, from small phytoplankton species (Neukermans et al., 2018) to larger organisms (Dalpadado et al., 2012). Recent studies have investigated the ecology and biodiversity of planktonic foraminifera from the high-latitude North Atlantic (e.g. Schiebel et al., 2017). The species *N. pachyderma* comprises more than 90% of recent assemblages (i.e. found in surface sediments) from the Polar Region, North of Iceland (Kucera et al., 2005). Rather few studies on living planktonic foraminifera (PF) communities have focussed on (sub-) Arctic regions. Plankton tows also show that *N. pachyderma* is dominant species in Arctic (~90%) followed by *T. quinqueloba* (5%) (Volkmann, 2000; Pados and Spielhagen, 2014). Through the compilation of population density profiles from 104 stratified plankton tow hauls collected in the Arctic and the North Atlantic Oceans, Greco et al. (2019) investigated the ecology of *N. pachyderma*. In particular, the variability of its habitat depth, and underlined the knowledge gap on its ecological preferences. In the western subpolar North Atlantic (Irminger Sea), the production of *N. pachyderma* shows two peaks, in spring and late summer, while winter shows a low production (Jonkers et al., 2010; 2013). The diversity of PF has increased in polar waters over the past decades, even though it remains low in comparison to lower latitudes, due to the poleward migration of warm-water species (Schiebel et al., 2017). A similar process appears to occur in the southern hemisphere (Meilland et al., 2016). Some species from lower latitudes are described as new components of poleward assemblages. The shift of PF assemblages to warmer conditions, since the pre-industrial stage, has been very recently highlighted more globally in the Northern hemisphere (Jonkers et al., 2019). These major modifications in PF distribution patterns display changes more related to primary production than to water temperature itself (e.g. Jonkers et al., 2010; Schiebel

et al., 2017). Planktonic foraminifera, being sensitive to ambient water geochemistry, are considered good indicators of the polar changing environments (Schiebel et al., 2017). More studies on living PF communities in the Arctic regions are needed to assess the spatial and temporal variability in their population dynamics and to better constrain the today's polar and subpolar species ecological preferences.

Taking the opportunity of a cruise dedicated to the exploration of the physical oceanography of the western Barents Sea (MOCOSED 2014 cruise), we investigated the connections between the spatial variability of living PF, phytoplankton communities (Giraudeau et al., 2016), and the hydrological system through a South-to-North transect, between Northern Norway and Spitsbergen [68-76]°N. Along this transect, we compared PF living faunas (from plankton tow) to the assemblages found on the sea floor (from core-top sediments) in order to investigate eventual recent changes in their population dynamics. This latitudinal transect also gave us the opportunity to quantify protein concentrations of individual living PF in this area for the first time and along a physico-chemical gradient to see if and how it varies and explore how planktonic foraminifera from a same species may adjust to different environments.

## 2. Oceanographic setting

The studied area covers the western Barents Sea margin, i.e. Barents Sea Opening (BSO), where the surface and intermediate ocean circulation are characterised by the confrontation of the North Atlantic and the Arctic Waters (Figure 1). The seasonal and interannual dynamics of these two water masses, interacting with the complex topography of the western margin (Storfjordrenna and Bjørnøyrenna glacial troughs separated by shallow Spitsbergenbanken), determine the position of the Polar Front (Loeng, 1991). The Norwegian Atlantic Current (NwAC) carries Atlantic Water into the Barents Sea. Along the western Barents Sea margin, Atlantic Water is transported to the Fram Strait by the West Spitsbergen current (Skagseth et al., 2008; Oziel et al., 2017; Figure 1).

In the southern part of the transect a strong thermohalocline clearly underlined a surface mixed layer of about 30-35 m depth. This cline slightly deepened northwards and blurred out north of 74.5°N where no stratification was observed in the water column. From South to North of the transect: i) high temperatures and low salinities reflected the Norwegian Coastal Water (NwCW), which is also enriched in Chl-*a*. The relatively warm NwCW (8.5 to 11°C) extended northwards up to 74.5°N overlying the colder Norwegian Atlantic Water (NwAW). Less saline (33.5) to the South, NwCW became saltier (34.9) in the vicinity of the Spitsbergenbanken; ii) at the northern end of the transect, the NwAW penetrated the Barents Sea through the Storfjordrenna trough with temperatures from 6 to 8°C and an open marine salinity of 35.1 (Giraudeau et al., 2016).

The Chl-*a* content followed the hydrological pattern described above (Figure 2 c). Relatively high concentrations (mean ≈ 0.8 mg.m$^{-3}$) were located in the surface mixed layer composed of NwCW. The highest values around 1.25 mg.m$^{-3}$ were recorded off the Norwegian coast. Chl-*a* content decreased northwards (north of 74.5°N) and reached ≈ 0.4 mg.m$^{-3}$ in the upper layer (0-60m) of the well-mixed NwAW. The composition of the phytoplankton community observed in surface water at 7 stations along the studied transect was essentially dominated by three algal groups: Fuco-flagelattes (25 to 43%; major component *Phaeocystis pouchetii*), Prasinophytes (15 to 30%; major components *Micromonas pusilla* and *Bathycoccus pusilla*) and

Prymnesiophytes (13 to 24%; major component *Emiliana huxleyi*) (Giraudeau at al., 2016). Three other features are noteworthy (Figure 2 d): i) the dominance of dinoflagellates (24%) at the southernmost station of the transect (close to the Norwegian coast) contrasted with their total absence in the well mixed NwAW, North of 74.5; ii) the presence of diatoms (10-20 %) in the surficial NwCW, but rare (<5%) to the North; iii) the constant increase in relative abundance of Cyanobacteria from < 5% to more than 15% along the South-to-North transect.

## 3. Material and Methods

In late summer 2014, from August 8 to September 20, the SHOM (French Hydrographic Office) operated the oceanographic cruise MOCOSED 2014, on board the "R/V *Pourquoi pas?*". Along a 700 km South-to-North transect from the Norwegian (68°N) to the Spitsbergen (76°N) coasts, investigations of hydrological processes at the BSO were carried out coupled with the exploration of the phytoplankton and foraminiferal communities using a total of 32 vertical casts deployed ≈ 20 km apart from each other (Figure 1 and 2).

### 3.1. Living planktonic foraminifera from stratified plankton samples (MultiNet)

Living PF were collected at 7 of the 32 CTD South-to-North transect stations (#3 to #9), and at 2 stations (#1 and #2) located West-to-East ≈ 20 km apart from the central point of the main South-to-North CTD transect (Figure 1; Table 1), using a stratified plankton tow (MultiNet Hydro-Bios type Midi, opening of 0.25 m²) equipped with five nets (mesh size 100 μm to avoid nets clogging in case of intense phytoplankton bloom). This sampling strategy was used in order to observe the potential effect of latitudinal changes but also of bathymetry, longitudinally, on PF distribution. At each station, one vertical haul sampled five successive water layers from the sea surface to 100 m depth. A second hauls has been deployed for the stations 1, 3, 4, 5, 6, 7 and 8, to collect material down to 700 m depth (Table 1). For each of the depth intervals (0–20 m, 20–40 m, 40–60 m, 60–80 m, 80–100 m; 0-100 m, 100-200 m, 200-300 m, 300-500 m and 500-700 m), the filtered water volume was measured by means of a flowmeter attached to the MultiNet mouth. Each MultiNet sample was preserved in a 250 mL vial with ethanol (90%) buffered with hexamethylenetetramine until processing at the land-based laboratory. Back at the laboratory, MultiNet samples were washed over a 100 μm mesh, all foraminifera were removed from the sample and dried in an oven at 50 °C. All living PF, distinguished by their coloured cytoplasm visible through the shell, were picked, stored in counting cells and identified at the species level, following the SCOR WG138 taxonomy as implemented in Siccha and Kucera (2017). Empty tests, considered as dead individuals were counted separately. Correlations following a non-metric multidimensional scaling ordination (NMDS) were carried out with the R package Vegan (Oksanen et al., 2013). Using the Bray-Curtis distance these correlations were tested between PF species absolute concentrations, the latitude of the station and parameters of the ambient waters (temperature, salinity, Chl-*a* concentration). Results are given in relative abundances (% of the total, live or dead fauna) or in absolute abundances in number of individuals per m³ of filtered water (ind.m⁻³).

*Protein biomass and test size measurements*

For protein extraction and measurement, a few living individuals (≈60) were picked on board from out of the shallowest water samples at stations 3 to 9 immediately after sampling. Only shells that were completely filled with cytoplasm were selected. After picking, individuals were immediately cleaned with a brush and filtered seawater to remove all particles, including organic matter, that were stuck to the test. The individuals were stored in a 1.5 mL Eppendorf vial and analysed on board, using the bicinchoninic acid (BCA) method as explained in Meilland et al., (2016). Morphometric analyses on single foraminiferal tests were carried out at the University of Angers with an automated incident light microscope (Bollmann et al., 2004; Clayton et al., 2009) at a resolution of 1.4 $\mu m^2$. Images were analysed for their two-dimensional (silhouette) morphometry (Beer et al., 2010), including minimum test diameter, which is the shortest distance from wall to wall that passes through the centre of the proloculus (the initial chamber of a foraminifer). Protein-to-size relations were determined for the minimum diameter of each test providing size-normalized protein content (SNP) for data analyses. Foraminifera protein concentrations were linearly normalized to 1 $\mu m$ minimum test diameter, being aware of any unavoidable errors related to non-linear increments of biomass at volumetric test growth (cf. Beer et al., 2010).

### 3.2. Fossil planktonic foraminifera assemblages from core-tops (Multitube)

At 5 sites of the main CTD transect, an interface corer (Multitube type Oktopus GmbH, Institut National des Sciences de l'Univers division of Brest, France) was used to obtain simultaneously 8 short sediment cores (less than 1 m in length) (Figure 1; Table 1). At each station, the core with the more even and undisturbed water-sediment interface was selected. The uppermost 0.5 cm of the core was sampled and fixed with 95 % ethanol. Samples were stained with Rose Bengal, reacting with cytoplasm, to distinguish PF still bearing cytoplasm (fresh or in degradation) and thus very recently deposited from fossil PF without cytoplasm. Stained shells of foraminifera probably reflect the spring and summer population of the year of sampling, even though the exact degradation time of cytoplasm is still poorly constrained (Schönfeld et al., 2013). The core-top sediments were wet sieved using a 100 $\mu m$ mesh and the foraminifera were identified using the same SCOR WG138 taxonomy in order to be directly comparable to the plankton tow samples.

### 4. Results

### 4.1. Planktonic foraminifera diversity and distribution in the water column

Data from the 7 stations of the South-to-North CTD transect with 5 depth values per station, were compiled to display the repartition of PF absolute abundances (for total assemblage and species-specific) in the upper 100 meters of the vertical section across the BSO. Abundances of living specimens below 100 m depth were very low (total concentration <5 ind.$m^{-3}$) and are therefore not discussed. Data from station 1 (Western) and station 2 (Eastern) located 20 km on either side of the S-N transect, were compared to the data of station 6 at the middle of the transect.

Total concentrations of living PF fauna varied between 0 and 400 ind.m$^{-3}$ (Figure 3). Along the South-to-North CTD transect, the highest concentrations were all observed above 20 m water depth, in the surface mixed layer of the well stratified water area, i.e. at the edge of the NwCW. The two stations located at the south and north extremities of the transect (# 3, off the Norwegian coast, and #9, off the Spitsbergen coast) displayed low densities (10 to 50 ind.m$^{-3}$). At station 1, located above the Barents Sea margin slope, the maximum abundance of 220 ind.m$^{-3}$ was recorded in a deeper habitat (20-40 m, Figure 3). Station 2, located inside the Barents Sea, was very poor in PF (<10 ind.m$^{-3}$). In total, 10 species were observed. The studied area was characterised by high abundances of subpolar to polar species (Figure 4), listed in descending order: *Globigerinita uvula* (45%), *Turborotalita quinqueloba* (26%), *Neogloboquadrina incompta* (15%) and *Neogloboquadrina pachyderma* (9%). There was a notable presence of the temperate water species *Globigerina bulloides* (3%), and negligible percentages (<1%) of *Globigerinita glutinata*, *Neogloboquadrina dutertrei*, *Globigerinoides ruber*, *Globigerinoides sacculifer* and *Orcadia riedeli*.

In the surface waters of the South-to-North transect (0-20 m depth), except at the septentrional station 9, *G. uvula* was the most abundant species reaching 64 % of the total fauna at station 7. The second most abundant species, *T. quinqueloba* dominated the PF fauna only at station 9, with 45 % of the total assemblage and 26 ind.m$^{-3}$. Both *G. uvula* and *T. quinqueloba* have a patchy repartition with two patches of maximum abundances located in the first 0-20 m, at station 4 and 7 for *G. uvula* (175 and 245 ind.m$^{-3}$, respectively) and at station 5 and 7 for *T. quinqueloba* (53 and 80 ind.m$^{-3}$, respectively). The northern and common patch for both species is the more intense. In the 20-40 m deep layer at station 1 (West of the transect), these two species showed also relatively high concentrations (121 ind.m$^{-3}$ for *G. uvula*, and 30 ind.m$^{-3}$ for *T. quinqueloba*). The two other major species *N. pachyderma* and *N. incompta* had similarly low relative abundances (4 to 22%, and 8 to 25%, respectively). For both species, maximum absolute abundance of about 40-45 ind.m$^{-3}$ occurred in the central part of the transect between [72-74°N].

The NMDS analysis of species abundances with regard to environmental parameters (latitude of the station, temperature, salinity, Chl-*a* concentration) indicates that none of the species-specific distribution displays a significant correlation to any of the tested variables (p-values > 0.1). NMDS documents distributional affinity (Figure 5), with *N. pachyderma* and *N. incompta* plotting in the same area and *T. quinqueloba* and *G. uvula* plotting separately from each and also from the *N. pachyderma*/*N. incompta* area.

## 4.2. Planktonic foraminifera protein biomass

Individual protein content (BCA method) and associated test minimum diameter were measured for a total of 272 specimens of the 4 major species, including 32 specimens of *Neogloboquadrina pachyderma*, 58 *Neogloboquadrina incompta*, 72 *Globigerinita uvula* and 110 *Turborotalita quinqueloba*. A 5 to 25 individuals per species were selected at each sampled depth-interval of the 7 stations along the S-N transect, paying careful attention to sample the whole size range of the populations. At

station 7, the protein extraction was successful for only one specimen of *N. pachyderma*. Therefore, no value is display for this species at this station in Figure 6.

Minimum diameters of the 272 selected tests cover a large size range, from 65 to 315 µm with a median value of 160 µm. *N. incompta* is the biggest species with a median of 200 µm, and *G. uvula* the smallest with a median of 110 µm (Table 2). For each studied species, the mean size is close enough to the median size to say that the size distribution of the picked tests is symmetric, thus making us confident that our test selection represents properly the natural test size range of each studied species. The biomass of a single individual normalized by its test size (SNP), averages out about 0.0055 µg of protein per µm

of foraminiferal shell diameter. It varies depending on species from 0.0004 (*G. uvula*) to 0.0426 µg.µm$^{-1}$ (*T. quinqueloba*). The SNP of all 4 species displays a northward decrease from 70 to 74° (Figure 6). *T. quiqueloba* and *G. uvula* have slightly (but not significantly) higher relative protein concentrations than *N. pachyderma* and *N. incompta* (Figure 6).

**4.3. Planktonic foraminifera diversity and distribution in surface sediments**

Concentrations of planktonic foraminifera with colourless empty tests varied from a maximum of 6200 ind.cm$^{-3}$ at station 4

(71.3°N) to a minimum of 200 ind.cm$^{-3}$ at the septentrional station 9 (Figure 7 a). *Neogloboquadrina pachyderma* was the most abundant species (31 to 59%) along the entire transect. Assemblages were more mixed at the two ends of the transect where *N. pachyderma* was less abundant. The assemblage at the southernmost point also contained *Turborotalita quinqueloba* (33%) and *Neogloboquadrina incompta* (24%). While at the northernmost point, station 9, *T. quinqueloba* (23%) co-occurred with *Globigerinita uvula* (25%).

Concentrations of planktonic foraminifera bearing a coloured cytoplasm (Figure 7 b) varied from 100 to 300 ind.cm$^{-3}$. All along the transect, the relative abundance of coloured *N. pachyderma* remained between 10 and 26 %. The species *T. quinqueloba* occurred everywhere above 20% and up to 40% South of 72°N. The central station 6 was dominated by *G. uvula* (38%). North of 74°, the fauna was balanced between *N. incompta* (33 and 9 %) and *G. uvula* (8 and 34%).

**5. Discussion**

*Distribution pattern of living planktonic foraminifera at the Barents Sea Opening*

In the late summer of 2014 the hydrology at the BSO was characterised by a strong water stratification with a 30 m thick Chl-*a* enriched lens of NwCW that to the north overlapped with the NwAW (saltier and colder) from 69.8°N to 74.5°N. Further north, a well-mixed water column with characteristics of the NwAW occupied the Storfjordrenna Trough. Here a coccolithophore bloom and the highest concentration of cyanobacteria were recorded in the upper water column (Giraudeau

et al., 2016). Despite these marked features the pattern of planktonic foraminifera abundance did not correlate with any of the studied environmental parameters (Figure 5). These observations confirm the low influence of commonly imputed parameters

such as temperature, salinity and primary production on PF density (Schiebel et al., 2017). In accordance with the conclusions of Retailleau et al., (2018) conclusions, multiples indications however suggest a possible role of water turbidity in PF abundance variation. The highest densities of PF occurred in the 0-20m upper water layer between 70.5 and 74.5°N. Their very low abundances (total concentration < 5 ind.m$^{-3}$) below 100 m depth suggest a shallow depth habitat for individuals in the region, especially for *N. pachyderma* which was recently reported to live between 25 and 280 m depth in the north Atlantic Arctic region (Greco et al., 2019). Very low abundances were also recorded nearby the Norwegian and Spitsbergen coasts. The low abundances at the two ends of the studied transect could reflect planktonic foraminifera patchiness pattern of distribution (Meilland et al., 2019) or highlight the fact that waters under continental influences (nutrient-enriched, more turbid) likely hamper the foraminiferal production. In line with this, the abrupt decrease in abundances from West to East (stations 2, to 6, to 1) may be ascribed to the decrease in depth of the Bjømøyrenna Trough up to the Barents Sea shelf (from 1850 to 430 m), as foraminifera are suspected to avoid neritic waters over continental shelves (Schmuker, 2000).

The remarkable point of our results is the dominance of *Globigerinita uvula*. This species, described as a temperate to polar species (Schiebel and Hemleben, 2017), is known to account for less than 2% of the assemblages in marginal Arctic Seas based on material collected with a 63 μm plankton net mesh size (Volkmann, 2000). *Neogloboquadrina pachyderma* is considered the dominant species in polar regions, making up more than 90% of the total planktonic foraminifera assemblages (e.g. Schiebel et al., 2017). The high densities of *G. uvula* recorded at the BSO in 2014 seem to contradict the former statements but are consistent with a recent study reporting *G. uvula* as one of the dominant species in southern high latitudes, South of the Polar Front (Meilland et al., 2017). A possible explanation could be the warming experienced by the western Barents Sea (SST anomalies ≈ +2°C) and its increase in salinity (SSS anomalies ≈ +0.3) over the last decades (Dobrynin and Pohlmann, 2015). These hydrological changes impact the plankton dynamics and biogeography, with a northward shift of the natural range of biological communities (Barton et al., 2016). Thus, the species distribution of planktonic foraminifera could be affected by an eventual expansion of subpolar/temperate species towards high latitudes leading to phytoplankton composition changes, in response to sea temperature warming under global climate change. Our observations from the North Polar Region support the shift of planktonic foraminifera assemblages to warmer conditions already asserted from North Atlantic (Jonkers et al., 2019) and from the southern Indian Ocean data (Meilland et al., 2017). However, a single observational dataset is the Barents Sea is not sufficient to robustly validate this assumption and a second hypothesis for the dominance of *G. uvula* in our sampling area could be a response to specific phytoplankton composition and ambient water conditions by pulsed reproduction events only in summer conditions. This seasonal pattern is known to occur in polar regions for *Turborotalita quinqueloba* (Schiebel and Hemleben, 2017). In fact, this species is the second dominant one in our late summer 2014 samples. As observed in this study, *T. quinqueloba* is also known to display high concentrations in the Barents Sea and western Spitsbergen (Volkmann 2000) and to co-occur with the typically polar species *Neogloboquadrina pachyderma* in the high-latitude cold-water assemblages (Volkmann, 2000; Eynaud, 2011).

Discrepancy between the species-specific distribution patterns was observed in late summer 2014 at the BSO. The low abundances of *Neogloboquadrina pachyderma* and *Neogloboquadrina incompta* consistent over the studied area versus the patchy distribution and high densities of *Globigerinita uvula* and *Turborotalita quinqueloba*, suggest differences in the ecological strategy and behaviour between these two pairs of species. The patchy pattern of planktonic foraminifera distribution has been observed before (Boltovskoy, 1971; Siccha et al., 2012; Meilland et al., 2019) suggesting that high densities are not exclusively constrained by the physical structure of the (sub-) surface layers.

Potential differences in diet preferences could explain the observed species distribution in late summer 2014 at the BSO. Both *G. uvula* and *T. quinqueloba* are thought to follow food availability and primary production (Volkmann 2000, Schiebel and Hemleben 2017). However, we observed no correlation between their distribution and Chl-*a* concentrations (Figure 5). In late summer 2014, *G. uvula* and *T. quinqueloba* showed high concentrations especially at station 7, located at the crossroads of the Atlantic (NwAW) and Arctic waters flowing out of the Storfjordrenna (Figure 1), at the edge of the polar front (Oziel et al., 2017). From this location to the northern station, the concentration of phytoplankton was relatively low and the phytoplankton community showed singular characteristics, in comparison to the southern part of the transect: fuco-flagelattes became dominant and diatom concentrations decreased. The fuco-flagelatte blooms (mainly *Phaeocystis pouchetii* in late summer 2014; Giraudeau et al., 2016) are well known to occur in the Barents Sea (Wassmann et al., 1990; Vaquer-Sunyer et al., 2013). Our hypothesis is thus that high densities of *G. uvula* and *T. quinqueloba* are due to food composition (quality) rather than food concentrations (quantity). This also implies that satellite-derived chlorophyll concentrations, considered as potential indicator of algal bloom, may not always be good indicators to perceive foraminiferal concentration and distribution.

*Planktonic foraminifera protein concentration, potential marker of their metabolism*

Proteins are the main component of zooplankton biomass ($C_{org}$) in all oceanographic regions, from the tropics to polar areas (e.g., Percy and Fife 1981; Donnelly et al., 1994; Kumar et al., 2013; Yun et al., 2015). Their role is essential to organisms' growth and their concentration and composition likely reflect the environment individuals grew in and how well they adjust to it. Based on previous studies, the protein concentration of PF can be used as a proxy of its biomass ($C_{org}$) and foraminiferal biomass should remain the same for a given test size (Schiebel and Movellan, 2012). However, in our study, the SNP (size normalized protein content) of PF decreases with higher latitude and hence with decreased in Chl-*a* concentration and temperature. The size of individuals picked for these analyses remains however constant. This observation suggests that foraminifera metabolisms (i.e. ability to consume/degrade food and to grow) is decreasing towards the north. This would be consistent with the observation of lower metabolism for zooplankton with decreasing salinity and temperature in the Arctic (Alcaraz et al., 2010). Proteins are the main component of zooplankton biomass ($C_{org}$), closely followed by lipids. Lipids in zooplankton organisms are very variable geographically, showing a latitudinal pattern with high percentages in polar areas and low percentages in warm tropical waters. Lipids percentages also display seasonal features, with higher values in summer than in winter (Falk-Petersen et al., 1999; Mayzaud et al., 2011; Kumar et al., 2013). It is thus possible that a part of energy (biomass

/ $C_{org}$) of the PF collected along the South-to-North transect shifts from being stocked as protein in warmer waters to being stocked as lipids in colder waters. This strategy would allow foraminifera to resist the cold to potentially overwinter. This hypothesis is supported by analogous observations made on different size fraction of zooplankton in the Southern Indian Ocean showing variability in protein and lipid percentages among the 80 to 200 µm populations (Harmelin-Vivien et al., 2019) and also by observations made on pteropods in the Arctic (Kattner et al., 1998; Phleger et al., 2001; Böer et al., 2005). The fact that higher SNP of foraminifera were observed where Chl-*a* is higher is compatible with the fact that polar organisms rely on their protein catabolism when food is easily accessible rather than on their lipid storage (Brockington & Clarke 2001). It has also been shown that a single organism in a cold environment is able to switch between predominantly protein or lipid catabolism across its life (Mayzaud, 1976). This suggests that individuals from the same species can display more or less proteins for the same biomass in different locations. With the reduction of PF protein concentration (and likely metabolism) going north, one could expect lower abundances. However, we observe no link between PF concentrations, which appeared to be species specific, and protein concentrations (evolving similarly for all four species) suggesting a decoupling between individuals metabolism and densities.

*Discrepancy between upper water column and interface sediment samples*

The PF species compositions recorded during the late summer 2014 in the water column and in surface sediments are similar while species relative abundances are drastically different. Indeed, the living fauna (collected by plankton net) displays large relative and absolute abundances of the two species *Globigerinita uvula* and *Turborotalita quinqueloba* whereas the fossil assemblages (found in core-tops) are largely dominated by *Neogloboquadrina pachyderma* or, at the southernmost station, co-dominated by *T. quinqueloba* and *N. pachyderma* (Figure 7 a). Affected by differential settling velocities (200 and 500 m.day$^{-1}$ in normal conditions), water depth and test sizes of different species, the foraminiferal fluxes exported from the upper productive surface and reaching the sea bottom depend on direction and intensity of currents (Takahashi and Bé, 1984). Lateral advection may transport shells over long distances > 25 km for *N. pachyderma* and > 50 km for *T. quinqueloba*, respectively (Von Gyldenfeldt et al., 2000). Lateral advection of shells is also strengthened by water stratification that increases resident time at the shear boundary between water masses (Kuhnt at al. 2013). The BSO has a complex hydrography with the buoyant NwCW flowing northwards above the NwAW that is entering eastwards the Barents Sea when cold BSAW is flowing westwards. In such areas, the PF settling velocities and extension of lateral advection poorly constrained. Consequently, the sediment core records cannot match exactly with the place and/or the intensity of production (Von Gyldenfeldt et al., 2000; Jonkers et al., 2015). However, considering potential lateral advection of shells, Pados and Spielhagen, (2014) observed from a study through the dynamic Fram Strait that the distribution pattern obtained by plankton tows was clearly reflected on the sediment surface, and that the assemblage on the sediment surface can be used as an indicator for modern planktonic foraminiferal fauna. This suggests that the large discrepancy between upper water column and interface sediment samples collected at the BSO in late summer 2014 should be taken into consideration. As a sedimentation rate of $1.3 \pm 0.6$ mm.yr$^{-1}$ has been recently measured in the Storfjordrenna outlet [76°N-17°E], close to our station 9 (Fossile et al., 2019), the core-top

sediments may have recorded less than a decade. Even though sedimentation rates are likely to vary along the transect, we hypothesise that the sea surface-bottom differences in the foraminiferal assemblages along the South-to-North transect at the BSO might reflect a community change within a short period of time (about a decade).

Furthermore, the analysis of sediment from the 5 core tops demonstrated important differences between the assemblages of
fossil fauna and recently settled tests (likely coming from surface Spring/Summer production), i.e. Rose-Bengal stained tests bearing not yet decomposed cytoplasm. For example, at 71.3°N, the percentages of coloured *T. quinqueloba* and *G. uvula* are twice as high as the ones observed for the fossil faunas (Figure 7). At 72.9°N in the surface sediment, *G. uvula* reaches up to 38% of the coloured assemblages (Figure 7 b) whereas it never exceeds 25% in the non-coloured ones (Figure 7 a). The high abundance of the two species *G. uvula* and *T. quinqueloba* in the living fauna as well as in the recently settled shells, but not
in the fossil faunas suggest that they may present a seasonal character with a production period focussed in late summer as a response to environmental and trophic conditions. This is supported by previous studies in the Arctic where *T. quinqueloba* has been found to dominate assemblages sampled in August (Carstens et al., 1997; Volkmann, 2000) but not in June/ early July (Pados and Spielhagen, 2014), and by sediment trap observations from the subpolar North Atlantic where *T. quinqueloba* reaches its maximum in autumn (Jonkers et al., 2010). The dominance of *N. pachyderma* in the fossil faunas collected at the
BSO and its low but constant presence in the coloured shells of surface sediment and plankton tow sampled in late summer 2014 suggests that this species may demonstrate a regular production throughout the year. *Neogloboquadrina pachyderma* production appears to be yearly sustained and constant in the area whereas other species clearly respond to a local seasonal signal.

**6. Conclusion**

The sampling and analytic approaches deployed during the MOCOSED14 cruise and combining the use of plankton net, core-top, molecular biology (protein measurement), environmental parameters and phytoplankton characterisation provides us with a unique dataset to better constrain the distribution of planktonic foraminifera within the highly complex studied area of the western Barents Sea.

The observed abundances of PF in the studied area are high offshore and the lower densities were recorded nearby the
Norwegian and Spitsbergen coasts. These observations highlight the fact that waters under continental influences (nutrient-enriched, more turbid) are rather inhospitable for PF production. The PF species composition observed at the BSO is diverse, with more than 10 different species in the net samples including *Globigerinita uvula* (45%), *Turborotalita quinqueloba* (26.2%), *Neogloboquadrina incompta* (15%) and *Neogloboquadrina pachyderma* (8.9%). The two species *G. uvula* and *T. quinqueloba* dominate the living (water sample) population and display highly patchy abundances suggesting they occur in
late summer in response to physico-chemical conditions and related specific primary productivity. The dominance of *G. uvula* in water samples could also be a signal of the temperature increase experienced over the last decades in the Barents Sea and

the North Atlantic Ocean. Further sampling in the area is thus needed to test this hypothesis. The species *N. pachyderma* and *N. incompta* show low densities but a continuous distribution pattern in the water samples. They also dominate the core-top assemblages suggesting that both species present a more consistent production over the course of spring-summer season.

Unlike their species-specific abundances pattern of distribution, size-normalized protein concentrations of all four major species decrease with the increasing latitude (and a decrease in temperature and Chl-*a* concentration). This observation leads us to hypothesise that 1) PF abundance and metabolism are decoupled and 2) foraminifera metabolism in the North of the studied region is lower than in the South. It opens the following question: Can individuals of the same species balance the ratio between their protein and lipid concentrations (major components of zooplankton $C_{org}$) in order to adapt to environmental

conditions (e.g. temperature)? Further analyses on planktonic foraminifera lipid concentration and composition are thus needed and would help us to better understand the metabolism of these organisms and their fate in a context of climate change.

**Data availability**

Data will be made available on request to the main author until their online publication on PANGAEA (https://pangaea.de/).

**Author contributions**

JM and HH designed the study. JM, VH, ID and JS generated the data and carried out the analyses. TG provided access to the MOCOSED 2014 cruise. All authors contributed to writing the manuscript.

**Competing interests**

The authors declare that they have no conflict of interest.

**Acknowledgements**

The authors are thankful to the crew and captain of R/V *Pourquoi pas ?*, as well as the scientific participants from the SHOM (French Hydrographic Office) for their support in sampling during the cruise "MOCOSED 2014". We fully acknowledge the efficient technical help provided by Sophie Sanchez at LPG-BIAF laboratory. We thank the three anonymous reviewers and Lukas Jonkers for their help on improving the manuscript. Grant was kindly provided to Vivien Hulot by the SHOM to study phytoplankton and PF communities.

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

**Table 1:** Location (Latitude and Longitude), sampling date and water depth, of the 9 MultiNet and 5 Multitube stations, incremented from South to North (stations 1 and 2 being positioned aside the main transect mid-point #6). Stations where phytoplankton analyses were performed are also indicated.

| Station | Latitude (°N) | Longitude (°E) | Date | Depth of sampling site (m) | Multinet (0-100 m) | Multinet (0-700 m) | Phytoplankton | Multitube |
|---|---|---|---|---|---|---|---|---|
| | | | | | Sample collection | | | |
| 3 | 69,845 | 13,879 | 22.08.14 | 2675 | x | x | x | x |
| 4 | 71,308 | 13,942 | 23.08.14 | 1940 | x | x | x | x |
| 5 | 72,138 | 14,098 | 23.08.14 | 1253 | x | x | x | |
| 1 | 72,893 | 11,762 | 16.08.14 | 1839 | x | x | | |
| 6 | 72,897 | 14,207 | 24.08.14 | 990 | x | x | x | x |
| 2 | 72,912 | 19,487 | 17.08.14 | 430 | x | | | |
| 7 | 73,736 | 14,376 | 24.08.14 | 1320 | x | x | x | |
| 8 | 74,537 | 14,508 | 25.08.14 | 1978 | x | x | x | x |
| 9 | 75,602 | 14,705 | 26.08.14 | 445 | x | | x | x |




**Table 2:** Descriptive statistics [minimum, maximum, average, and median values in μm] of the minimum (i.e. the shortest) diameter of all the 272 BCA measured specimens, per species.

| Species | Test size distribution parameters in μm | | | |
|---|---|---|---|---|
| | Minimum | Average | Median | Maximum |
| All | 65.12 | 158.45 | 160.2 | 315.18 |
| *G. uvula* | 65.5 | 108.8 | 108.8 | 160.8 |
| *T. quinqueloba* | 65.12 | 166.13 | 166.56 | 249.18 |
| *N. incompta* | 75.08 | 191.29 | 196.66 | 280.46 |
| *N. pachyderma* | 100.7 | 184.3 | 180.7 | 315.18 |

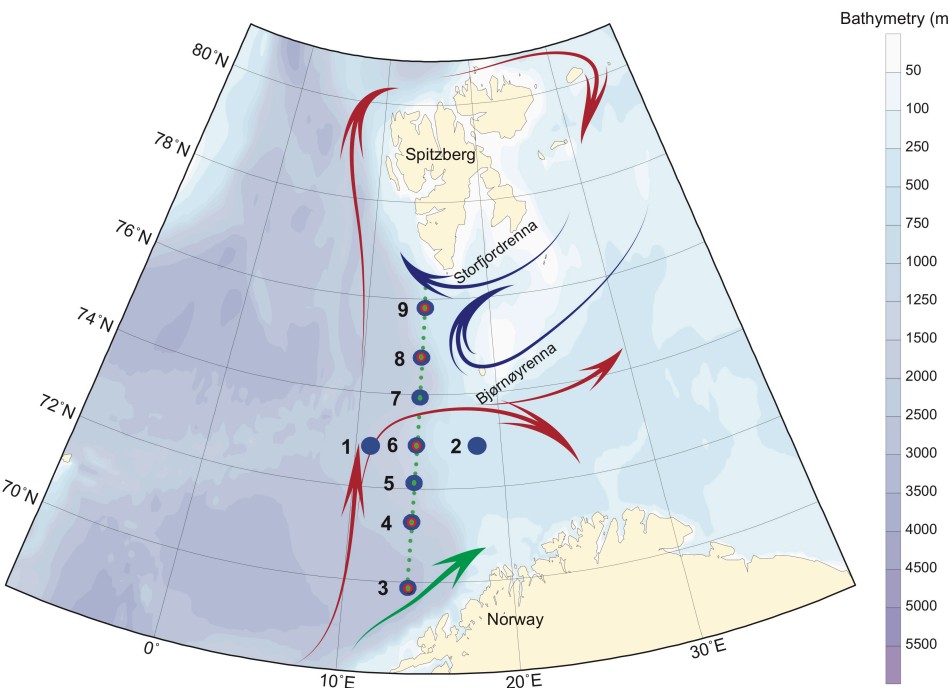


**Figure 1:** Sampling map of the MOCOSED cruise in the western Barents Sea with schematic surface circulation (red arrows = Atlantic Water; blue arrows = Arctic Water; green arrow = Norwegian Coastal Current; adapted from Oziel et al., 2017). Little green dots display the 32 CTD/Niskin stations along the South-to-North transect; large blue circles show the location of the 9 MultiNet stations; medium red circles underline the 5 multicore stations.


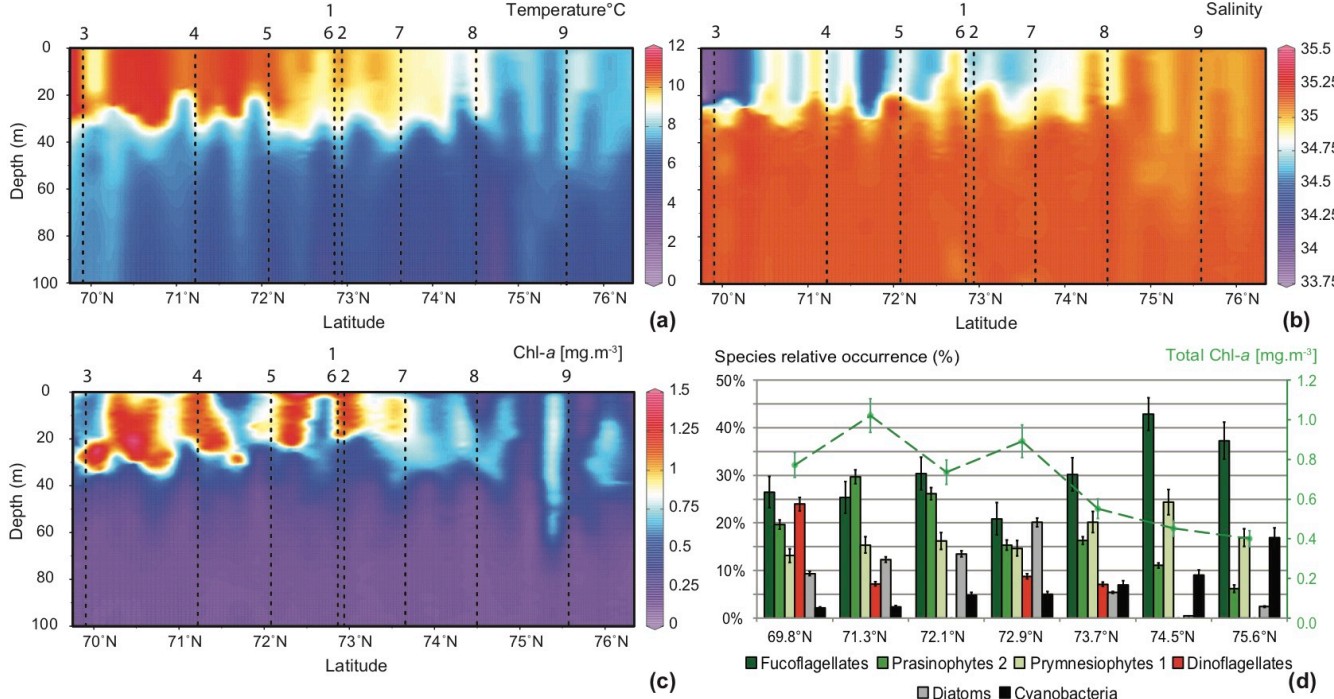

**Figure 2:** South-North, 0-100 m deep sections across the Barents Sea Opening, compiled from data of the 32 CTD casts: (a) Temperature (°C); (b) Salinity; and (c) Chl-*a* concentrations (mg.m⁻³). Vertical dashed lines and associated numbers correspond to the location of the MultiNet hauls. The lower right panel (d) displays pigment concentrations translated into the relative abundances of the major phytoplankton species at 7 sampling stations along the transect (Giraudeau et al., 2016).

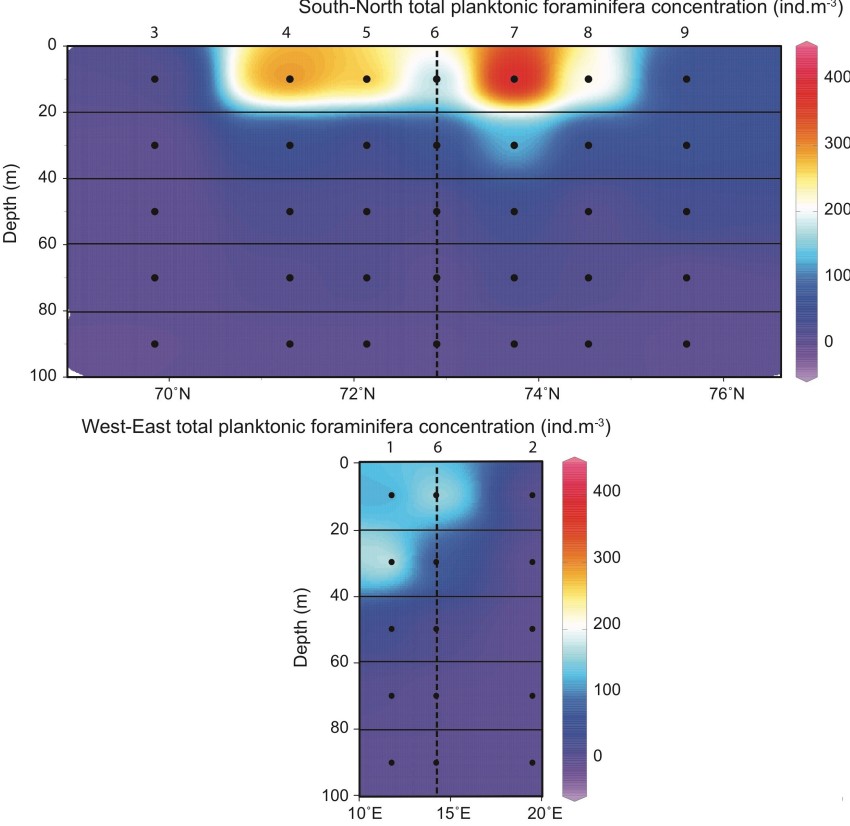

**Figure 3:** Distribution of planktonic foraminifera total abundances (ind.m⁻³) in the 0-100 m depth section across the Barents Sea Opening (upper panel) and in the West-East transect (stations 1, 6 and 2; lower panel). Station names are indicated above vertical alignments of 5 dots representing the middle points of the 5 net sampling intervals.


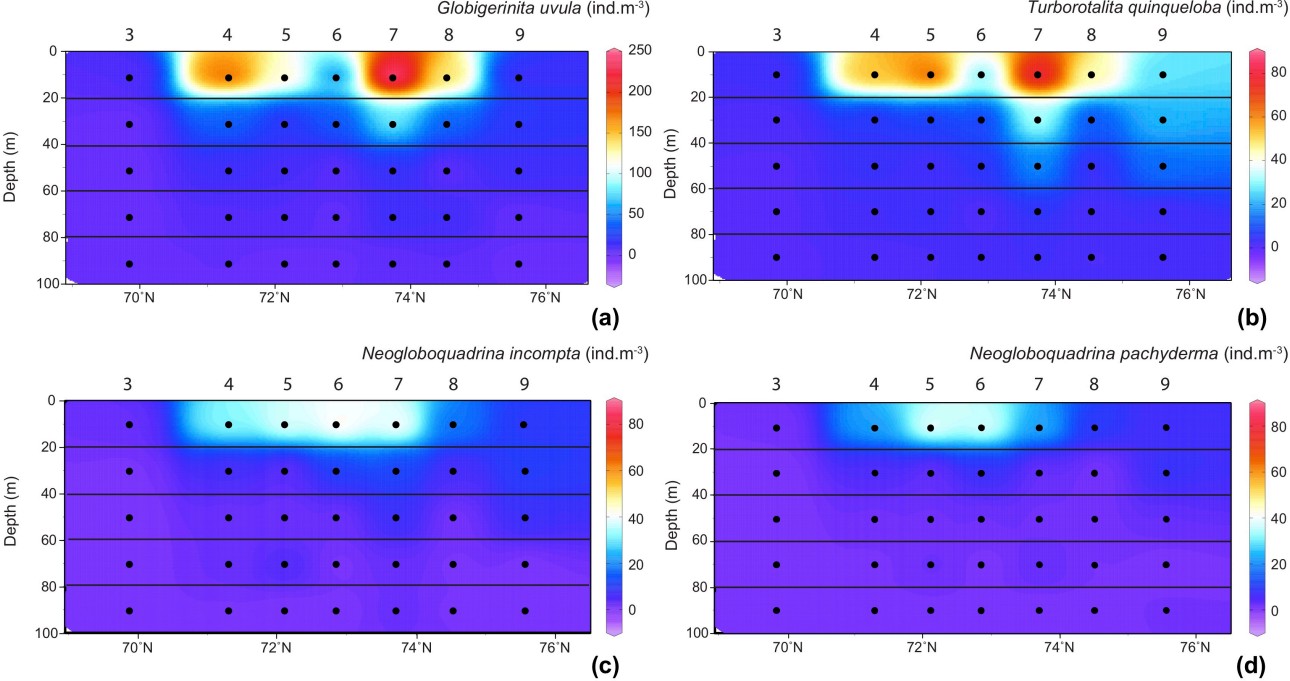

**Figure 4:** Distribution of the four major species abundances (ind.m$^{-3}$) in the 0-100 m depth section across the Barents Sea Opening. (a) For *Globigerinita uvula* with species abundances (z-axes) going from 0 to 250 ind.m$^{-3}$; (b) for *Turborotalita quinqueloba* (c) *Neogloboquadrina incompta* (d) and *Neogloboquadrina pachyderma* values goes from 0 to 80 ind.m$^{-3}$. Station names are indicated above vertical alignments of 5 dots representing the middle points of the 5 net sampling intervals.

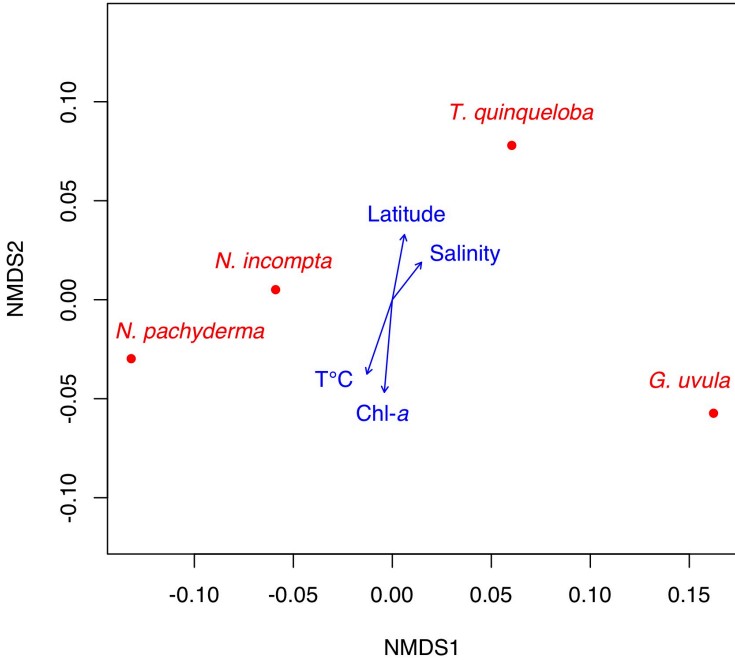


**Figure 5:** Standard nonmetric multidimensional scaling ordination analysis (NMDS) of planktonic foraminifera species distribution (red dots) with temperature, salinity, Chl-*a* and station location as factor (blue arrows).

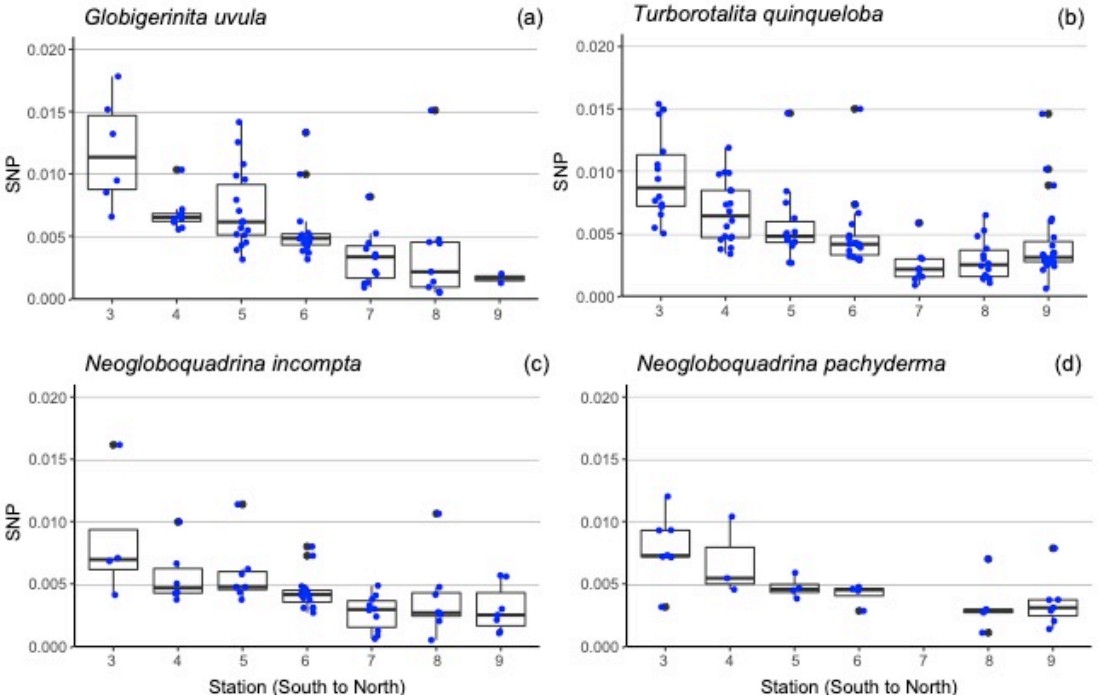


**Figure 6:** Boxplot of the size normalized protein biomass (SNP, µg.µm$^{-1}$), along the South-to-North transect (from station 3 = 69.8°N to station 9 = 75.6°N) (a) for *Globigerinita uvula* (b), *Turborotalita quinqueloba* (c), *Neogloboquadrina incompta* (d) and *Neogloboquadrina pachyderma*. Blue dots highlight the data dispersion. Potential outliers were removed such as data for *N. pachyderma* at station 7 as analyses were only run on one individual. Thick lines indicate median, boxes extend to

interquartile range (IQR) and whiskers indicate 1.5*IQR.

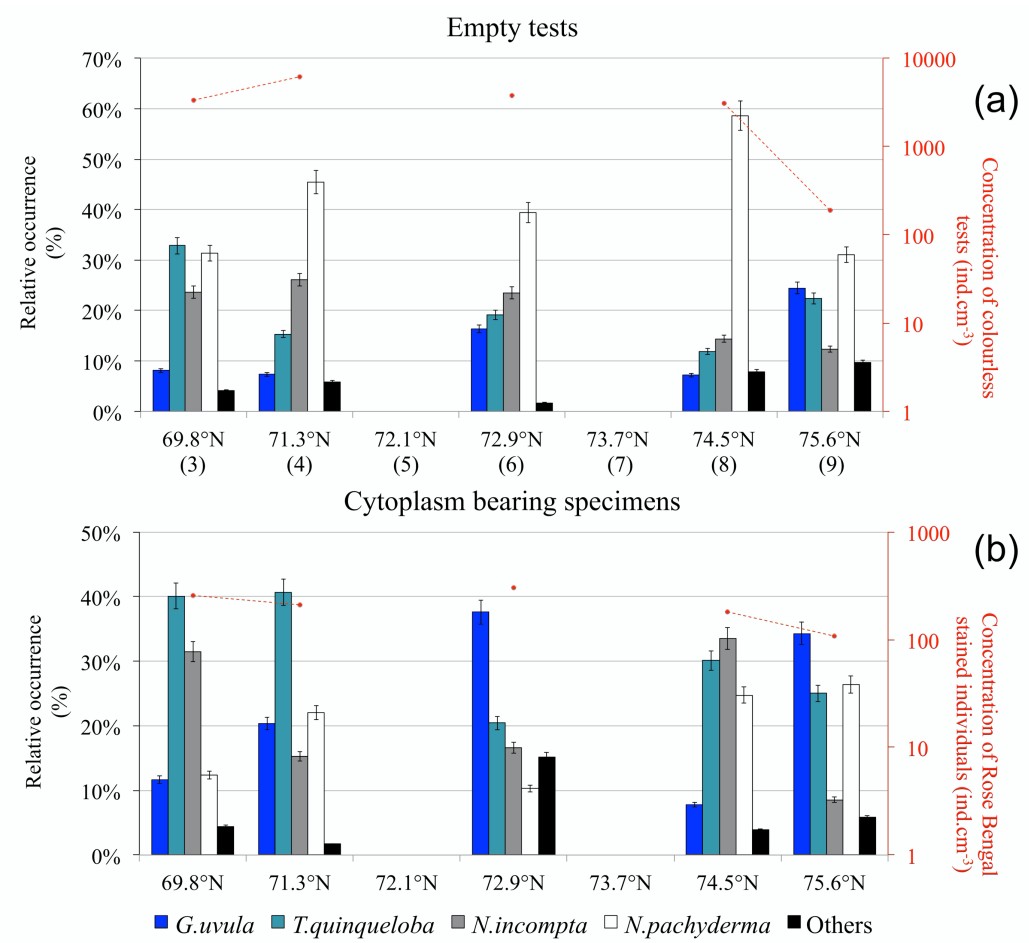

**Figure 7:** Relative species occurrence (% of the total fauna) of planktonic foraminifera found in the upper 0.5 cm core-top sediment (histograms) and total concentration of individuals per cm$^3$ of dry sediment (logarithmic Y axis on the right): (a) colourless empty tests (stations numbers are in brackets) and (b) individuals bearing a Rose Bengal coloured cytoplasm.