# Peer review of "Population dynamics of modern planktonic foraminifera in the western Barents Sea"

_Biogeosciences, 2019_

## Referee Comment (RC1) · Anonymous Referee #1 · 25 Nov 2019

This paper compare plankton town and sediment cores to gain insight in the PF population dynamics in the Western Barent Sea. Protein analysis is also performed on foraminifera test as proxy of metabolism. The authors can cut few sentences in the introduction which will be benefit in fluency. Some methodology information are within the results session and need to be removed/merged within the methods session. Also it is not clear wherever or not Chla and phytoplankton data are new data or already published in another paper. This clarification will imply some change within over the text. Most important, the discussion presents several strong statements which need to be better supported to reduce the amount of speculation. The conclusion have to be re-write in order to be a critical synthesis of the paper and not just a summary of the paper. Overall, the combination of data is interesting and the paper merit a publication on this journal but after medium revision. Please refers to specific comments for more details: INTRODUCTION Lines 40-44 This part in not fluent and need to be re-organise/shorted. The authors first talk about phytoplankton compositions, then they list calcifying organisms (including zooplankton). There is also no need to highlight the non-calcifying organisms since it not help the reader to focus on the main question the paper want to address. Line 49-51 As for the previous comment, there is no need to add more information about fish community. This sentence reduce the fluency of the paper. I suggest to delete it. Line 56: before to use the abbreviation PF, the author need to identify what this means. Planktonic foraminifera (PF). Please be consistent over the all text. Line 71-74: Move the sentence "planktonic foraminifera..indicator..changing environments" before the sentence " more studies on living...ecological preferences" Line 75-76: This information appears in the text 3 time: introduction, methods and acknowledgment. Please remove from the introduction. Line 79: What living fauna is referring to? I assume PF but wrote in this way looks like the full zooplankton assemblage. Line 80-83 This sentence need to be re-write because it is a bit confused as it is presented. LPF individual protein are investigated from net samples. However here seems like protein have been analysed also in core samples. The author need also to take more effort in describe why it is relevant to do this study in the Barent Sea and why it is relevant to do protein analysis. In other word the author need to work a bit more on how to "set the scene" Methodology 3.1 My understanding is that phytoplankton analysis (pigment and composition) are coming from a previous study. If it is this the case, the authors should not include this information in the method and neither in the results. Line 110 I am aware the fraction smaller than 63micron can be relatively low, however the author should acknowledge somehow the decision to use 100micron instead 63micron. 3.3 Not quite understand the reason to have a different head line. 3.3 is presenting analysis of protein from forams collected in the net. It is much more fluent to have only 3 headlines in the methodology i-hydrological environmental collection, ii-town, iii-core. So in this case will be sufficient just to merge 3.2 with 3.3. Results 4.1 As for my previous comments this session have to be removed since it is not a result

of Meilland et al. I understand that the author will use this data to compare with forams data. This is fine but need to be part of the discussion only. 177-180: This information need to be moved/merged in the methods and the reason of the selection of the 2 transect have to be clarify better. Line 178: Does the author means total and relative abundance? Please be consistent along the text with the terminology Line 208-214: Most of this information need also to be moved/merged in the methods Line 225-228: as for previous comment please move/merge with methods Discussion: Line 262-263 The author investigate the possibility of the high abundance of GU as potential consequence of the climate change. This is a big statement supported only by data collected in one single shot (not time series) in the ocean. I suggest either remove or at least to acknowledge the limitation of this statement. Also this statement is in disagreement with what the author said before (lines 246-247) about the low influence of T and S on PF density. Please clarify better. What about the potential influence of net mesh size? Can the small missing fraction bias the relative abundance between species? What about timing in collection (day/night)? Are the author assuming the foraminifera do not perform diel vertical migration? If this is the case need to be supported by literature. Line 284-294 Do the author found a specific correlation between Phaeocystis and GU or TQ in all the stations? Also the author have to explain better why Phaeocystis is considered high quality food, why they should prefer it? What is the strategy diet difference between GU, TQ and NP? The author need to expand this part to better defend the statement. Line 294 please provide literature of study which use Chl-a satellite data as indicator of foraminifer's extension production as example. Line 298-316 the discussion linked to the protein results is a bit disconnected with the rest. Can protein results help the authors to drown a better picture concerning the relation/discrepancy between net versus cores? Despite the variability of protein concentration with the latitude is an intriguing result it does add to much to the value of the paper in the way it is included in the discussion. Line 340 Similar to previous comment. Can the authors really speculate that a collection of a sample in a specific time can be indicative of a shift in population when compared a decade average from the sediment core? Speculation

is allowed in a certain perimeter but it is very important that the authors acknowledge and clarify the limitation of their statement. Conclusion The conclusion need to be re-organize and shorted. In general the conclusion should be not just a summary. To me this looks more like a summary at the end of thesis chapters than a conclusion. The authors have to provide a synthesis of the results in order to highlight the relevance of them within a big pictures. This is a relative short paper so there is no need to re call point by point (from a to f!) all the results achieved. What the author need to provide here is a critical thinking and elaboration of the most relevant MS findings and what are the new insight they bring in the marine research community.

---

## Referee Comment (RC2) · Anonymous Referee #2 · 25 Nov 2019

The paper by Meiland et al. presents a really interesting study of planktonic foraminiferal occurrence in the Barents Sea. Notably, the study includes both plankton tow and core-top samples, including Rose Bengal staining of recently deposited foraminifera. They have also included an analysis of protein biomass in their methodology, which could be an interesting complement to their observations. Overall, the study is quite interesting, however, I have some suggestions for potentially improving analysis and presentation, which I hope the authors will consider.

Overarching comments: 1) It appears to me that one of the critical limitations of the study at presented is a lack of time constraints. The authors compare planktonic foraminifera standing stock (instantaneous), to Rose Bengal stained (integrated over weeks to... years?), and unstained core tops (integrated over decades?). The comparison between abundances across these different timescales is potentially a huge strength of the work, but it is difficult to interpret without further time constraints and/or clear discussion of these issues. Could the authors, for example: a. Include an estimate or discussion of what Rose Bengal stained sediment top foraminifera represent? A month? A season? A year? b. Timing between and dates of plankton tows? It looks as if these tows were taken over the course of ∼6 weeks. If so, this should be made explicit, with dates included, and discussed. Especially as the authors discuss both seasonal and lunar production in some species, this is a potentially important point. Could assemblages have changes of the course of late summer to fall? Are different periods in the lunar cycle being sampled? c. Include a more thorough discussion of the evidence for a temperature-driven change in assemblage over the past decades? While I agree this is hypothetically plausible, given the uncertainties in timescales outlined above and the well-described seasonality and patchiness of planktonic foraminifera in tows, this is not currently a convincing line of argument based on the data.

2) The inclusion of protein biomass measurements is a particularly interesting aspect of this work, but the results are not clearly synthesized in the discussion. For example, I'm struggling to understand how conclusion e) that planktonic foraminiferal dynamics and metabolism are decoupled, relates to the data. I'd urge the authors to be more explicit first about how protein biomass is a proxy for metabolism, and then be very specific in discussing what aspects of "dynamics" and metabolism are decoupled. My confusion may stem from lack of expertise in this area, but clarifying the importance of these findings and linking them to the conclusions can only increase the impact for a less specialized audience.

This comment is obviously stylistic, but I would discourage the overuse of acronyms to improve overall readability. For example there is no need to abbreviate "planktonic foraminifera" to PF. Additionally, if acronyms must be used, please avoid starting sentences with them, ie., the second sentence of the abstract.

17: Subfossil -> just say core top if you mean core top

18: four same -> same four

29: is -> are

42: exhibit -> exhibiting

47: is -> are

71: no "highly"

73: no "as"

125: "a few"

126: CTD -> CTD

128: how were foraminifera cleaned?

143: I don't think this is correct. Rose Bengal staining should indicate the presence of organic material, but gives no information about the presence of coloured cytoplasm.

155-156: This sentence requires some clarification

Section 4.2. Can you be consistent with the significant digits on the foraminiferal relative abundances?

241: where in the "South"?

247: to -> with

248-255: I am wary of the over-interpretation of these results given that they are based on single tows and the repeated observations of planktonic foraminiferal patchiness (including as discussed in this paper and in Meiland et al., 2019).

259: no "as"

267: remove "best probably"

299-301: please clarify that "size" refers to shell size, not cell size. Is it possible that

part of what you are observing could be a decoupling of shell and cell sizes at high latitudes?

---

## Author Response (AR1)

Authors: We thank the editor, Carol Robinson for taking the time to go through our manuscript and comments to the reviewers. We amended the manuscript following the two referees comments and we hope this new version of it will satisfy you.
Our answers to the reviewers are written in red and highlighted in the "tracked" version of the manuscript (below) in green.

Authors: We thank the Referee#1 for taking time to review our manuscript and appreciate the valuable comments and suggestions. We have addressed the comments in the following sections and in the revised manuscript:

This paper compare plankton town and sediment cores to gain insight in the PF population dynamics in the Western Barent Sea. Protein analysis is also performed on foraminifera test as proxy of metabolism. The authors can cut few sentences in the introduction which will be benefit in fluency.

Authors: We followed reviewer's comment and modified the introduction accordingly.

Some methodology information are within the results session and need to be removed/merged within the methods session. Also it is not clear wherever or not Chla and phytoplankton data are new data or already published in another paper. This clarification will imply some change within over the text.

Authors: We followed reviewer's suggestions and re-organized the text accordingly.

Most important, the discussion presents several strong statements which need to be better supported to reduce the amount of speculation. The conclusion have to be re-write in order to be a critical synthesis of the paper and not just a summary of the paper. Overall, the combination of data is interesting and the paper merit a publication on this journal but after medium revision. Please refers to specific comments for more details.

Authors: some of the statements in the discussion have been toned down (e.g. L.239, 257-258) and more details were added when needed. We re-arranged and shortened the conclusions.

INTRODUCTION
Lines 40-44 This part in not fluent and need to be reorganise/shorted. The authors first talk about phytoplankton compositions, then they list calcifying organisms (including zooplankton). There is also no need to highlight the non-calcifying organisms since it not help the reader to focus on the main question the paper want to address.

Authors: We agree with the reviewer and we shortened and re-wrote this part.

Line 49-51 As for the previous comment, there is no need to add more information about fish community. This sentence reduce the fluency of the paper. I suggest to delete it.

Authors: The sentence has been delete.

Line 56: before to use the abbreviation PF, the author need to identify what this means. Planktonic foraminifera (PF). Please be consistent over the all text.

Authors: Changes were made L.53 and over the manuscript.

Line 71-74: Move the sentence "planktonic foraminifera..indicator..changing environments" before the sentence " more studies on living. . .ecological preferences"

Authors: Done

Line 75-76: This information appears in the text 3 time: introduction, methods and acknowledgment. Please remove from the introduction.

Authors: Done

Line 79: What living fauna is referring to? I assume PF but wrote in this way looks like the full zooplankton assemblage.

Authors: We amended the sentence according to the reviewer's comment L.75.

Line 80-83 This sentence need to be re-write because it is a bit confused as it is presented. LPF individual protein are investigated from net samples. However here seems like protein have been analysed also in core samples. The author need also to take more effort in describe why it is relevant to do this study in the Barent Sea and why it is relevant to do protein analysis. In other word the author need to work a bit more on how to "set the scene"

Authors: The sentence has been re-written and more details given about the relevance of protein quantification. Protein represents a large part of zooplankton organic carbon composition and could provide crucial informations on individuals' food availability, uptake and ecological strategy. Doing these measures in our study area is extremely relevant as 1) the studied latitudinal transect allows to observe a wide range of S and T°C and thus potential adaptation of foraminifera and 2) as no data are available in the region.

Methodology 3.1 My understanding is that phytoplankton analysis (pigment and composition) are coming from a previous study. If it is this the case, the authors should not include this information in the method and neither in the results.

Authors: All information relative to environmental parameters were shifted to the "oceanographic setting" section of the paper as indeed, previously published by Giraudeau et al., 2016.

Line 110 I am aware the fraction smaller than 63micron can be relatively low, however the author should acknowledge somehow the decision to use 100micron instead 63micron.

Authors: The sampling occurred in summer/fall 2014 when phytoplankton blooms are known to occur. To avoid clogging in the nets and because it is a very standard mesh-size globally (in mid- low- latitudes), we choose to use a mesh size of 100 μm. We acknowledged this decision manuscript L.118.

3.3 Not quite understand the reason to have a different head line. 3.3 is presenting analysis of protein from forams collected in the net. It is much more fluent to have only 3 headlines in the methodology i-hydrological environmental collection, ii-town, iii-core. So in this case will be sufficient just to merge 3.2 with 3.3.

Authors: We agree with the reviewer and merged sections together.

Results 4.1
As for my previous comments this session have to be removed since it is not a result of Meilland et al. I understand that the author will use this data to compare with forams data. This is fine but need to be part of the discussion only.

Authors: This section has been shifted under "Oceanographic settings" from L.87 to L.108.

177-180: This information need to be moved/merged in the methods and the reason of the selection of the 2 transect have to be clarify better.

Authors: We understand the reviewer's comment however this short paragraph is used here to help the reader and we wish to keep it there. The choice of the two studied transect is now justified in the Material and Methods section L.115 to L.119.

Line 178: Does the author means total and relative abundance? Please be consistent along the text with the terminology

Authors: We make the distinction between absolute abundance (ind. m$^{-3}$) and relative abundance (%). We don't mention total abundances in the manuscript or when we do it is to refer to all species together.

Line 208-214: Most of this information need also to be moved/merged in the methods

Authors: The fact that protein measurements were successful on 272 specimens is already a result and we think it belongs to this part. The complementary information of this paragraph are here to help reader going through this section.

Line 225-228: as for previous comment please move/merge with methods

Authors: We understand the reviewer's comment however this short paragraph is used here for the context and help the reader. We would prefer to keep it there.

Discussion
Line 262-263 The author investigate the possibility of the high abundance of GU as potential consequence of the climate change. This is a big statement supported only by data collected in one single shot (not time series) in the ocean. I suggest either remove or at least to acknowledge the limitation of this statement.

Authors: As suggested by the reviewer, we toned down our statement and acknowledged its limitations in the manuscript.

Also this statement is in disagreement with what the author said before (lines 246-247) about the low influence of T and S on PF density. Please clarify better.

Authors: (previous) Lines 246-247 concerned the potential influence of environmental parameters on the density of planktonic foraminifera as a population while the lines the reviewer refers to concerns the density/ecology of one species in particular: *G. uvula*. This is why we separated the two paragraphs.

What about the potential influence of net mesh size? Can the small missing fraction bias the relative abundance between species?

Authors: The influence of the used mesh size would bias results the other way around with more specimens of small size (*G. uvula* and *T. quinqueloba*). Therefore the fact that we used a 100µm mesh size only support the fact that the observation of *G. uvula* and *T.quinqueloba* in such densities is surprising.

What about timing in collection (day/night)? Are the author assuming the foraminifera do not perform diel vertical migration? If this is the case need to be supported by literature.

Authors: We indeed assume planktonic foraminifera do not perform diel vertical migration, as published by Meilland et al., 2019, cited in the manuscript.

Line 284-294 Do the author found a specific correlation between Phaeocystis and GU or TQ in all the stations? Also the author have to explain better why Phaeocystis is considered high quality food, why they should prefer it? What is the strategy diet difference between GU, TQ and NP? The author need to expand this part to better defend the statement.

Authors: In this paragraph (L. 271 to 283) we only speculate on the fact that *G. uvula* and *T. quinqueloba* could reflect food composition more than food availability. We do not comment on the nutritional quality of Phaeocystis. To answer the second part of the reviewer's comment, only little is known about in situ diet preferences for these species, especially in the studied region. It is therefore difficult to go further in our hypothesis.

Line 294 please provide literature of study which use Chl-a satellite data as indicator of foraminifer's extension production as example.

**Authors: Here we say Chl-a satellite data are used for the observation of phytoplankton bloom, not necessarily as indicator of PF extension. However, several studies compare Chl-*a* satellite data to PF distribution it is for example the case in one of our previous publication (Meilland et al., 2016).**

Line 298-316 the discussion linked to the protein results is a bit disconnected with the rest. Can protein results help the authors to drown a better picture concerning the relation/discrepancy between net versus cores? Despite the variability of protein concentration with the latitude is an intriguing result it does add to much to the value of the paper in the way it is included in the discussion.

**Authors: Based on both reviewer's comments we provided more information about the relevance of protein analyses and re-wrote parts of this section. (L.285 to 288).**

Line 340 Similar to previous comment. Can the authors really speculate that a collection of a sample in a specific time can be indicative of a shift in population when compared a decade average from the sediment core? Speculation is allowed in a certain perimeter but it is very important that the authors acknowledge and clarify the limitation of their statement.

**Authors: We fully agree with the referee's comment and we toned down our statement.**

Conclusion
The conclusion need to be reorganize and shorted. In general the conclusion should be not just a summary. To me this looks more like a summary at the end of thesis chapters than a conclusion. The authors have to provide a synthesis of the results in order to highlight the relevance of them within a big pictures. This is a relative short paper so there is no need to re call point by point (from a to f!) all the results achieved. What the author need to provide here is a critical thinking and elaboration of the most relevant MS findings and what are the new insight they bring in the marine research community.

**Authors: the conclusion has been completely reorganized and shortened following the reviewer's suggestions.**

**Authors: We thank the Referee#2 for taking time to review our manuscript and appreciate the valuable comments and suggestions. We have addressed the comments in the following sections and in the revised manuscript:**

Referee#2: The paper by Meiland et al. presents a really interesting study of planktonic foraminiferal occurrence in the Barents Sea. Notably, the study includes both plankton tow and core-top samples, including Rose Bengal staining of recently deposited foraminifera. They have also included an analysis of protein biomass in their methodology, which could be an interesting complement to their observations. Overall, the study is quite interesting, however, I have some suggestions for potentially improving analysis and presentation, which I hope the authors will consider.

Overarching comments:

1) It appears to me that one of the critical limitations of the study at presented is a lack of time constraints. The authors compare planktonic foraminifera standing stock (instantaneous), to Rose Bengal stained (integrated over weeks to... years?), and unstained core tops (integrated over decades?). The comparison between abundances across these different timescales is potentially a huge strength of the work, but it is difficult to interpret without further time constraints and/or clear discussion of these issues.

**Authors: We agree with Reviewer's comment and provided additional information (L.154 to 156) about the different time constraints. Based on literature and sedimentation rate in the study area (Fossile et al., 2019), we can safely say the unstained core tops represent less than a decade.**

Could the authors, for example:

a. Include an estimate or discussion of what Rose Bengal stained sediment top foraminifera represent? A month? A season? A year?

Authors: We can reasonably think that the stained organisms represent the Spring/summer population that recently felt and the discussion is based on this assumption. We amended the Material and Methods accordingly to make it clear fo the reader (L.151). The sediments in the studied area are oxidised and therefore it is safe to say Rose Bengal wouldn't stain foraminifera over a long time period (i.e. a year).

b. Timing between and dates of plankton tows? It looks as if these tows were taken over the course of 6 weeks. If so, this should be made explicit, with dates included, and discussed. Especially as the authors discuss both seasonal and lunar production in some species, this is a potentially important point. Could assemblages have changes of the course of late summer to fall? Are different periods in the lunar cycle being sampled?

Authors: Sampling dates are available in Table 1. The latitudinal transect (South to North) was sampled across 4 days only in August. We can therefore not expect a change of assemblages in the water column due to the summer/fall transition or to the lunar cycle.

c. Include a more thorough discussion of the evidence for a temperature-driven change in assemblage over the past decades? While I agree this is hypothetically plausible, given the uncertainties in timescales outlined above and the well-described seasonality and patchiness of planktonic foraminifera in tows, this is not currently a convincing line of argument based on the data.

Authors: We understand reviewer's concern and we toned down this hypothesis over the manuscript.

2) The inclusion of protein biomass measurements is a particularly interesting aspect of this work, but the results are not clearly synthesized in the discussion. For example, I'm struggling to understand how conclusion e) that planktonic foraminiferal dynamics and metabolism are decoupled, relates to the data.
I'd urge the authors to be more explicit first about how protein biomass is a proxy for metabolism, and then be very specific in discussing what aspects of "dynamics" and metabolism are decoupled. My confusion may stem from lack of expertise in this area, but clarifying the importance of these findings and linking them to the conclusions can only increase the impact for a less specialized audience.

Authors: Proteins are one of the main constituents of organic carbon in living organisms and come from the food organisms consume and transform (metabolise). They support organisms' growth and give us information on how they feed: are organisms starving? Adapting to/degrading food? We can therefore suspect that for individuals of a same size a reduction in protein concentration could be the signal of a metabolism slow down. This can be explain by 1) a lack of resources, 2) unsuitable resources, 3) a global unsuitable environment, 4) a change of "behaviour" (i.e. dormance) etc… If foraminifera have a lower metabolism in the North of the studied area, one could expect to observe fewer individuals but our observations do not show a link between abundance of foraminifera and their protein concentration. That's the decoupling we are talking about.
We added more details about the relevance of protein quantification over the manuscript (e.g. L.285 to 288) and we clarified our message on the decoupling between foraminifera dynamics (abundances) and metabolism (protein concentrations).

This comment is obviously stylistic, but I would discourage the overuse of acronyms to improve overall readability. For example there is no need to abbreviate "planktonic foraminifera" to PF. Additionally, if acronyms must be used, please avoid starting sentences with them, ie., the second sentence of the abstract.

Authors: We understand the Reviewer's remark and amended the abstract accordingly.

17: Subfossil -> just say core top if you mean core top
Authors: we made the change L.17.

18: four same -> same four
Authors: we made the change L.18.

29: is -> are
Authors: we made the change L.29.

42: exhibit -> exhibiting
Authors: we made the change L.41.

47: is -> are
Authors: we made the change L.46.

71: no "highly"
Authors: we removed it.

73: no "as"
Authors: we removed it.

125: "a few"
Authors: we made the correction L.133.

126: CTD -> CTD
Authors: we made the correction L.134.

128: how were foraminifera cleaned?
Authors: Foraminifera were cleaned with a brush and filtered sea water. We amended the text accordingly L.136.

143: I don't think this is correct. Rose Bengal staining should indicate the presence of organic material, but gives no information about the presence of coloured cytoplasm.
Authors: We amended the sentence L.150-151 accordingly "This organic stain reacting with cytoplasm was used here to distinguish PF still bearing fresh cytoplasm and thus very recently deposited from empty tests of fossil PF"

155-156: This sentence requires some clarification
Authors: We agree and finally decided to remove the sentence.

Section 4.2. Can you be consistent with the significant digits on the foraminiferal relative abundances?
Authors: We checked the consistency of digits and corrected them when needed (e.g. L.175).

241: where in the "South"?
Authors: South has been replaced by "69.8°N" L.228.

247: to -> with
Authors: We made the correction L.234.

248-255: I am wary of the over-interpretation of these results given that they are based on single tows and the repeated observations of planktonic foraminiferal patchiness (including as discussed in this paper and in Meiland et al., 2019).
Authors: We agree with the reviewer and mentioned patchiness as a plausible explanation "The low abundances at the two ends of the studied transect could reflect planktonic foraminifera patchiness pattern of distribution (Meilland et al., 2019) or highlight the fact that waters under continental influences…" L.238 – 239.

259: no "as"

Authors: we removed it.

267: remove "best probably"
Authors: we removed it.

299-301: please clarify that "size" refers to shell size, not cell size. Is it possible that part of what you are observing could be a decoupling of shell and cell sizes at high latitudes?
Authors: Following the reviewer's comment we added "test" before size L.291. A strong enough decoupling of shell and cell sizes at high latitudes to explain our observations seems unlikely since specimens selected for protein measurements were selected on the basis of cytoplasm presence in all visible chambers. Also, none of the specimens were encrusted and the "available" space in the shell should have therefore been comparable.

**Population dynamics of modern planktonic foraminifera in the western Barents Sea**

Julie Meilland[1], Hélène Howa[2], Vivien Hulot[3,4], Isaline Demangel[5,6], Joëlle Salaün[7], Thierry Garlan[7]

[1] MARUM - Center for Marine Environmental Sciences, Leobener Str. 8, D-28359, Bremen, Germany

[2] LPG-BIAF, UMR-CNRS 6112; University of Angers, France

[3] University of French Polynesia, UMR-241 EIO, Labex Corail, FAA'A, Tahiti, French Polynesia

[4] Ifremer, UMR-241 EIO, Labex Corail, Departement Ressources Biologiques et Environnement, Vairao, Tahiti, French Polynesia

[5] Institute of Earth Sciences, University of Graz, NAWI Graz Geocenter, Graz, Austria

[6] Department of Geology, University of Lund, Lund, Sweden

[7] SHOM –Sciences et Techniques Marines/Géologie Marine, Brest, France

*Correspondence to*: Julie Meilland (jmeilland@marum.de)

**Abstract.** This study reports on species diversity and distribution of planktonic foraminifera (PF) at the Barents Sea Opening (BSO). Populations of PF living in late summer (collected by means of stratified plankton tows) and recently settled individuals (sampled by interface corer) were studied and compared. High abundances reaching up to 400 ind.m$^{-3}$ in tow samples and 8000 ind.cm$^{-3}$ in surface sediments were recorded in the centre of the studied area while low abundances were observed in coastal areas, likely hampered by continental influences. The living and core-top assemblages are mainly composed of the same four species *Neogloboquadrina pachyderma, Neogloboquadrina incompta*, *Turborotalita quiqueloba* and *Globigerinita uvula*. The two species *G. uvula* and *T. quiqueloba* largely dominate the upper water column whereas surface sediment assemblages display especially high concentrations of *N. pachyderma*. The unusual dominance of *G. uvula* in the water sample assemblages compared to its low occurrence in surface sediments might be the signature of 1) a seasonal signal due to summer phytoplankton composition changes at the BSO, linked to the increase of summer temperature at the study site, and/or 2) a signal of a larger time-scale and wide geographical reach phenomenon inducing poleward temperate/subpolar species migration and consecutive foraminiferal assemblage diversification at high latitudes under global climate forcing. Protein concentrations were measured on single specimens and used as a proxy of individual carbon biomass. Specimens of all species show the same trend, i.e. a northward decrease of their size-normalized-protein concentration suggesting foraminiferal biomass to be potentially controlled by different constituents of their organelles (e.g. lipids). The originality of coupling data from plankton tows, protein measurements and surface sediments allows us to hypothesise that PF dynamics (seasonality and distribution) are decoupled from their metabolism.

**Keywords:** living and dead communities, latitudinal distribution, protein content, seasonality, atlantification

**1 Introduction**

Polar areas are sensitive to global temperature changes, particularly in the Arctic where warming occurs faster than in the rest of the world and has accelerated over the past 50 years (Shepherd, 2016). This Arctic amplification appears to be mainly caused by sea-ice loss under increasing $CO_2$ (Dai et al., 2019). Recent increased advection of Atlantic Water in the Barents Sea modifies its physico-chemical properties (Smedsrud et al., 2013), which gets directly reflected in the entire ecology of the region. Higher temperatures lead to increased rates of planktonic primary production (Vaquer-Sunyer et al., 2013) and increased $CO_2$ concentrations are expected to have a fertilization effect on marine autotrophs (Holding et al., 2015). Enhanced primary production is accompanied by lateral shifts of the spring and summer phytoplankton blooms in the European Arctic Ocean (Oziel et al., 2017). As a response some taxa of different calcifying groups (i.e. foraminifera, coccolithophores, molluscs and echinoderms; Beaugrand et al., 2013) exhibit a poleward movement in agreement with expected biogeographical changes under sea temperature warming. Both satellite images (Smyth et al., 2004; Burenkov et al., 2011) and *in situ* measurements (Dylmer et al., 2013; Giraudeau et al., 2016; AMAP 2018) have recorded rapid expansion of temperate species of coccolithophores in the Arctic. For example, *Emiliania huxleyi* shows a striking poleward shift (>5°) in the distribution of its blooms (Neukermans et al., 2018). Such phenomenon, called "atlantification" (Årthun et al., 2012), are expected to impact every trophic levels of the food web, from small phytoplanktonic species (Neukermans et al., 2018) to larger organisms (Dalpadado et al., 2012). Recent studies have investigated the ecology and biodiversity of planktonic foraminifera from the high-latitude North Atlantic (i.e. Schiebel et al., 2017). Eynaud, (2011) noticed that the species *N. pachyderma*, the most characteristic high-latitude taxon, dominates the past interglacial assemblages for the last 1.8 Ma. while the subpolar species *Turborotalia quinqueloba* records northward penetration of Atlantic warm water masses in the Arctic, especially during interglacial periods. The species *N. pachyderma* comprises more than 90% of recent assemblages (i.e. found in surface sediments) from the Polar Region, North of Iceland (MARGO data base; Kucera et al., 2005). Rather few studies on living planktonic foraminifera (PF) communities have concentrated on (sub-) Arctic regions. Pados and Spielhagen (2014) observed PF by the means of plankton tows, in the cold and fresh Polar waters of the Fram Strait during early summer. They report a large dominance of *N. pachyderma* and a co-occurrence of *T. quinqueloba*, accounting for 90 and 5% of all tests, respectively. Volkmann, (2000) also documented this large dominance of *N. pachyderma* and the co-occurrence with *T. quinqueloba,* overall in the Arctic. Through the compilation of population density profiles from 104 stratified plankton tow hauls collected in the Arctic and the North Atlantic Oceans, Greco et al., (2019) deeply investigated the ecology of *N. pachyderma*. In particular, the variability of its habitat depth, and finally underlined the knowledge gap on its ecological preferences. In the western subpolar North Atlantic (Irminger Sea), the maximum production of *N. pachyderma* shows two peaks, in spring and late summer, while winter shows a low production (Jonkers et al., 2010; 2013). Following the extensive review of Schiebel et al., (2017), diversity of planktonic foraminifera has increased

in polar waters over the past decades, even though it remains low in comparison to lower latitudes. Some species from lower latitudes are described as new components of formerly high-latitude assemblages (Southern Indian Ocean; Meilland et al., 2016). The shift of planktonic foraminifera assemblages to warmer conditions, since the pre-industrial stage, has been very recently highlighted more globally in the Northern hemisphere (Jonkers et al., 2019). These major modifications in PF distribution patterns display changes more in primary production than in water temperature itself (e.g. Jonkers et al., 2010; Schiebel et al., 2017). Planktonic foraminifera, being sensitive to ambient water geochemistry, are considered good indicators of the polar changing environments (Schiebel et al., 2017). More studies on living PF communities in the Arctic regions are needed to assess the spatial and temporal variability in their population dynamics and to better constrain the today's polar and subpolar species ecological preferences.

Taking the opportunity of a cruise dedicated to the exploration of the physical oceanography of the western Barents Sea (MOCOSED 2014 cruise), we investigated the connections between the spatial variability of living planktonic foraminifera, phytoplankton communities (Giraudeau et al., 2016), and the hydrological system through a South-to-North transect, between Northern Norway and Spitsbergen [68-76]°N. Along this transect, we compared PF living faunas (from plankton tow) to the assemblages found on the sea floor (from core-top sediments) in order to investigate eventual recent changes in their population dynamics. This latitudinal transect also gave us the opportunity to quantify protein concentrations of individual living PF in this area for the first time and along a physico-chemical gradient to see if and how it varies and explore how planktonic foraminifera from a same species may adjust to different environments.

**2. Oceanographic setting**

The studied area covers the western Barents Sea margin, i.e. Barents Sea Opening (BSO), where surface and intermediate ocean circulations are characterised by the confrontation of the North Atlantic and the Arctic Waters (Figure 1). The seasonal and interannual dynamics of these two water masses, interacting with the complex topography of the western margin (Spitsbergen Banken and shallow Bjørnøya; Storfjordrenna and Bjømøyrenna glacial troughs), determine position and meandering of the Polar Front (Loeng, 1991). The Norwegian Atlantic Current (NwAC) carries Atlantic Water into the Barents Sea. Along the western Barents Sea margin, Atlantic Water is then transported to the Fram Strait by the West Spitsbergen current (Skagseth et al., 2008; Oziel et al., 2017; Figure 1).

Environmental parameters (temperature, salinity and fluorescence) as well as phytoplankton composition were obtained during the MOCOSED 2014 cruise using a total of 32 vertical casts deployed ≈ 20 km apart from each other (Figure 1 and 2, Giraudeau et al., 2016). In the southern first half of the transect a strong thermohalocline clearly underlined a surface mixed layer of about 30-35 m depth. This cline slightly deepened northwards and blurred out north of 74.5°N where no more stratification was observed in the water column. From South to North of the transect: i) high temperatures and low salinities reflected the Norwegian Coastal Water (NwCW), also enriched in Chl-*a*. The relatively warm NwCW (8.5 to 11°C) extended northwards up to 74.5°N overlying the colder Norwegian Atlantic Water (NwAW). Less saline (33.5) to the South, NwCW became saltier (34.9) in the vicinity of the Spitsbergen Banken; ii) at the northern end of the transect, the NwAW

penetrated the Barents Sea through the Storfjordrenna trough with temperatures from 6 to 8°C and an open marine salinity of 35.1.

The Chl-*a* content followed the hydrological pattern above described (Figure 2 c). Relatively high concentrations (mean ≈ 0.8 mg.m$^{-3}$) were located in the surface mixed layer composed of NwCW. The highest values around 1.25 mg.m$^{-3}$ were recorded off the Norwegian coast. Chl-*a* content decreased northwards (north of 74.5°N) to reach ≈ 0.4 mg.m$^{-3}$ in the upper layer (0-60m) of the well-mixed NwAW. The composition of the phytoplankton community observed in surface water at 7 stations along the studied transect was essentially dominated by three algal groups (Giraudeau at al., 2016): Fuco-flagelattes (25 to 43%; major component *Phaeocystis pouchetii*), Prasinophytes (15 to 30%; major components *Micromonas pusilla* and *Bathycoccus pusilla*) and Prymnesiophytes (13 to 24%; major component *Emiliana huxleyi*). Three other features are noteworthy (Figure 2 d): i) the dominance of dinoflagelattes (24%) at the southernmost station of the transect (close to the Norwegian coast) contrasted with its total absence in the well mixed NwAW, North of 74.5; ii) the presence of diatoms (10-20 %) in the surficial NwCW, but rare (<5%) to the North; iii) the constant increase in relative abundance of Cyanobacteria from < 5% to more than 15%, along the South-to-North transect.

**3. Material and Methods**

In late summer 2014 from August 8 to September 20, the SHOM (French Hydrographic Office) operated the oceanographic cruise MOCOSED 2014, on board the "R/V *Pourquoi pas ?*". Along a 700 km South-to-North transect from the Norwegian (68°N) to the Spitzberg (76°N) coasts, investigations of hydrological processes at the BSO were carried out coupled with the exploration of the phytoplankton and foraminiferal communities (Figure 1).

**3.1. Living planktonic foraminifera from stratified plankton samples (MultiNet)**

Living PF were collected at 7 of the 32 CTD South-to-North transect stations (#3 to #9), and at 2 stations (#1 and #2) located West-to-East ≈ 20 km apart from the central point of the main South-to-North CTD transect (Figure 1; Table 1), using a stratified plankton tow (MultiNet Hydro-Bios type Midi, opening of 0.25 m2) equipped with five nets (mesh size 100 μm to avoid nets clogging in case of intense phytoplankton bloom). This collection was set in order to observe the potential effect of latitudinal changes but also of bathymetry, longitudinally, on PF distribution. At each station, one single vertical haul sampled five successive water layers from the sea surface to 100 m depth. For each of the five depth intervals (0–20 m, 20–40 m, 40–60 m, 60–80 m, 80–100 m), the filtered water volume was measured by means of a flowmeter attached to the MultiNet mouth. Each MultiNet sample was preserved in a 250 mL vial with ethanol (90%) buffered with hexamethylenetetramine until processing at the land-based laboratory. Back at the laboratory, MultiNet samples were washed over a 100 μm mesh, all foraminifera were removed from the sample and dried in an oven at 50 °C. All living PF, distinguished by their coloured cytoplasm visible through the shell, were individually picked, stored in counting cells and identified at the species level, following the SCOR WG138 taxonomy as implemented in Siccha and Kucera (2017).

Correlations following a non-metric multidimensional scaling ordination (NMDS) were carried out with the R package Vegan (Oksanen et al., 2013). Using the Bray-Curtis distance these correlations were tested between PF species absolute abundances, the latitude of the station and parameters of the ambient waters (temperature, salinity, Chl-*a* concentration).

130 Empty tests, considered as dead individuals were separately numbered. Results are given in relative abundances (% of the total, live or dead fauna) or in absolute abundances in number of individuals per $m^3$ of filtered water ($ind.m^{-3}$).

*Protein biomass and test size measurements*

Immediately after sampling (on board), a few living individuals ($\approx$60) were picked out of the shallowest water samples (0 – 20 m) of the 7 stations sampled along the main CTD transect (stations 3 to 9) for protein extraction and measurement. Each

135 foraminifer picked for protein measurement was carefully selected under the strict condition of its shell to be fully filled with cytoplasm. After picking, individuals were immediately cleaned with a brush and filtered seawater to remove all particles stuck to the test including organic matter. Individual were stored in a 1.5 mL Eppendorf cup and analysed on board, using the bicinchoninic acid (BCA) method as explained in Meilland et al., (2016). Morphometric analyses on single foraminiferal tests were carried out at the University of Angers with an automated incident light microscope (Bollmann et al., 2004;

140 Clayton et al., 2009) at a resolution of 1.4 $\mu m^2$ (pixel size). Images were analysed for their two dimensional (silhouette) morphometry (Beer et al., 2010), including foraminiferal test minimum diameter being the shortest distance wall to wall passing through the centre of the proloculus (the initial chamber of a foraminifer). Protein-to-size relations were determined for the minimum diameter of each test providing size-normalized protein content (SNP) for data analyses and handling. Foraminifera protein concentrations were linearly normalized to 1 $\mu m$ minimum test diameter, being aware of any

145 unavoidable errors related to non-linear increments of biomass at volumetric test growth (cf. Beer et al., 2010).

**3.2. Fossil planktonic foraminifera assemblages from core-tops (Multitube)**

At 5 sites of the main CTD transect, an interface corer (Multitube type Oktopus GmbH, INSU[1] division of Brest, France) was implemented to obtain simultaneously 8 short sediment cores (less than 1 m in length) (Figure 1; Table 1). At each station, the core with the more horizontal and undisturbed water-sediment interface was selected. The core-top sediment (0-

150 0.5 cm slice) was sampled and fixed with 95 % ethanol and Rose Bengal. This organic stain reacting with cytoplasm was used here to distinguish PF still bearing cytoplasm (fresh or in degradation) and thus very recently deposited from empty tests of fossil PF. The last study on Rose Bengal-stained individuals of Foraminifera, focussing only on benthic ones did not remove the uncertainty about the exact duration of complete cytoplasm degradation in the tests (Schönfeld et al., 2013), thus we cannot be precise on the time-scale pointed out by Rose Bengal-stained specimens. Based on the hydrology, sites depth

155 (Table 1), and sediment oxidation over the studied area, we can reasonably think that Rose Bengal coloration is in our situation highlighting spring and summer population that recently felt. The discussion will be based on this assumption.
* * *
[1] INSU : Institut National des Sciences de l'Univers

For the purpose of this study, the core-top sediment has been wet sieved on a 100 µm mesh (same as the plankton net mesh size), and analysed for the planktonic foraminiferal assemblages. Every picked planktonic foraminifer has been identified consistently with those collected by plankton tows.

**4. Results**

**4.1. Planktonic foraminifera diversity and distribution in the water column**

[revised manuscript text omitted]

Interface undisturbed cores were retrieved from 5 stations of the South-to-North CTD transect (Table 1) to investigate the core-top sediment (0-0.5 cm slice). The dead PF assemblages were studied making the difference between Rose Bengal-stained showing recently dead individuals still bearing a non-degraded cytoplasm after post-mortem deposition, and colourless empty tests of individuals dead for longer periods of time.

Concentrations of planktonic foraminifera with colourless empty tests (Figure 7 a) varied from a maximum of 6200 ind.cm$^{-3}$ at station 4 (71.3°N) to a minimum of 200 ind.cm$^{-3}$ at the septentrional station 9. All along the S-N transect, the fossil

assemblages were dominated by *Neogloboquadrina pachyderma* (31 to 59%). Assemblages were more balanced at the two ends of the transect where *N. pachyderma* showed off at its lowest occurrence. At the southernmost point, station 3 presents a co-occurrence with *Turborotalita quinqueloba* (33%) and *Neogloboquadrina incompta* (24%). While at the northernmost point, station 9, *T. quinqueloba* (23%) co-occurred with *Globigerinita uvula* (25%). Concentrations of planktonic foraminifera bearing a coloured cytoplasm (Figure 7 b) varied from 100 to 300 ind.cm$^{-3}$. All along the transect, the relative abundance of *N. pachyderma* remained between 10 and 26 %. The species *T. quinqueloba* occurred everywhere above 20% and up to 40% South of 72°N. The central station 6 was largely dominated by *G. uvula* (38%). North of 74°, the fauna was balanced between *N. incompta* (33 and 9 %) and *G. uvula* (8 and 34%).

**5. Discussion**

*Distribution pattern of living planktonic foraminifera at the Barents Sea Opening*

In late summer 2014 the hydrology at the BSO was characterised, from 69.8°N to 74.5°N, by a strong water stratification with a 30 m thick Chl-*a* enriched lens of NwCW overlapping northwards the NwAW (saltier and colder). Further North, a well-mixed water column with characteristics of the NwAW occupied the Storfjordrenna trough where a coccolithophore bloom (Giraudeau et al., 2016) and the highest concentration of cyanobacteria were recorded in the upper water column. Despite these marked features the global pattern of planktonic foraminifera abundance did not correlate with any of the studied environmental parameters (Figure 5). These observations confirm the low influence of commonly imputed parameters such as temperature, salinity and primary production on PF density (Schiebel et al., 2017). In accordance with Retailleau et al., (2018) conclusions, multiples indices however highlight the possible importance of water turbidity in PF abundance distribution. The highest densities of planktonic foraminifera occurred in the 0-20m upper water layer between 70.5 and 74.5°N, and very low abundances were recorded nearby the Norwegian and Spitzbergen coasts. The low abundances at the two ends of the studied transect could reflect planktonic foraminifera patchiness pattern of distribution (Meilland et al., 2019) or highlight the fact that waters under continental influences (nutrient-enriched, more turbid) likely hamper the foraminiferal production. In line with this, the abrupt decrease in abundances from West to East (stations 2, to 6, to 1) may be ascribed to the decrease in depth of the Bjømøyrenna trough up to the Barents Sea shelf (from 1850 to 430 m), as foraminifera are suspected to avoid neritic waters over continental shelves (Schmuker, 2000).

The remarkable point of our results is the dominance of *Globigerinita uvula* in the high-latitude (> 70°N) waters at the BSO. This species, described as a temperate to polar species (Schiebel and Hemleben, 2017), is known to occupy less than 2% of the assemblages in marginal Arctic Seas based on material collected with a 63 µm plankton net mesh size (Volkmann, 2000). *Neogloboquadrina pachyderma* is considered the dominant species in polar regions, making up more than 90% of the total planktonic foraminifera assemblages (e.g. Schiebel et al., 2017). The high densities of *G. uvula* recorded at the BSO in 2014 seem to be inconsistent with the former statements but are consistent with a recent study reporting *G. uvula* as one of the

dominant species in southern high latitudes, South of the Polar Front (Meilland et al., 2017). A possible explanation to these observations could be the warming experienced by the western Barents Sea (SST anomalies ≈ +2°C) and its increase in salinity (SSS anomalies ≈ +0.3) over the last decades (Dobrynin and Pohlmann, 2015). These hydrological changes impact the plankton dynamics and biogeography, with a northwards shift of the natural range of biological communities (Barton et al., 2016). Thus the species distribution of planktonic foraminifera could be affected by an eventual expansion of subpolar/temperate species towards high latitudes leading to phytoplankton composition changes, in response to sea temperature warming under global climate change. Our observations from the North Polar Region support the shift of planktonic foraminifera assemblages to warmer conditions already asserted from North Atlantic (Jonkers et al., 2019) and from the southern Indian Ocean data (Meilland et al., 2017). However, a single observational dataset is the Barents Sea is not sufficient to robustly validate this assumption and a second hypothesis for the dominance of *G. uvula* in our sampling area could be a response to specific phytoplankton composition and ambient water conditions by pulsed reproduction events only in summer conditions. This seasonal pattern is known to occur in polar regions for *Turborotalita quinqueloba* (Schiebel and Hemleben, 2017). In fact, this species is the second dominant one in our late summer 2014 samples. As observed in this study, *T. quinqueloba* is also known to display high concentrations in the Barents Sea and western Spitsbergen (Volkmann 2000) and to co-occur with the typically polar species *Neogloboquadrina pachyderma* in the high-latitude cold-water assemblages (Volkmann, 2000; Eynaud, 2011).

Discrepancy between the species-specific distribution patterns was observed in late summer 2014 at the BSO. The low abundances of *Neogloboquadrina pachyderma* and *Neogloboquadrina incompta* consistent over the studied area versus the patchy distribution and high densities of *Globigerinita uvula* and *Turborotalita quinqueloba*, suggest differences in the ecological strategy and behaviour between these two pairs of species. The patchy pattern of planktonic foraminifera distribution has been observed before (Boltovskoy, 1971; Siccha et al., 2012; Meilland et al., 2019) suggesting that high densities are not exclusively constrained by the physical structure of the (sub-) surface layers.

Potential differences in diet preferences could explain the observed species distribution in late summer 2014 at the BSO. Both *G. uvula* and *T. quinqueloba* are supposed to follow food availability and primary production (Volkmann 2000, Schiebel and Hemleben 2017), but we did not observe any correlation between their distribution and Chl-*a* concentrations (Figure 5). In late summer 2014, *G. uvula* and *T. quinqueloba* showed high concentrations especially at station 8, located at the cross road of the Atlantic (NwAW) and Arctic waters flowing out of the Storfjordrenna (Figure 1), at the edge of the polar front (Oziel et al., 2017). For this particular location, the concentration of phytoplankton was relatively low and the phytoplankton community showed singular characteristics, in comparison to the southern part of the studied transect: fuco-flagelattes became dominant and diatom concentrations decreased. The fuco-flagelatte blooms (mainly *Phaeocystis pouchetii* in late summer 2014; Giraudeau et al., 2016) are well known to occur in the Barents Sea (Wassmann et al., 1990; Vaquer-Sunyer et al., 2013). Our hypothesis thus is that *G. uvula* and *T. quinqueloba* high densities reflect more food composition

(quality) than food concentrations (quantity). This also implies that satellite-derived chlorophyll concentrations, considered as indices for algal bloom, may not always be good indicators to perceive neither lateral extension nor intensity of foraminiferal production.

[revised manuscript text omitted]

Furthermore, the analysis of sediment from the 5 core-tops collected during the MOCOSED cruise demonstrated important differences between the assemblages of fossil fauna, i.e. empty tests of individuals dead for a while, and recently settled tests
340 (likely coming from surface Spring/Summer production), i.e. Rose-Bengal stained tests bearing not yet decomposed cytoplasm. For example, at 71.3°N, the percentages of coloured *T. quinqueloba* and *G. uvula* are twice higher than the ones observed for the fossil faunas (Figure 7). At 72.9°N in the surficial sediment, *G. uvula* reaches up to 38% of the coloured assemblages (Figure 7 b) whereas it never exceeds 25% in the non-coloured ones (Figure 7 a). The large representation of the two species *G. uvula* and *T. quinqueloba* in the living fauna as well as in the recently settled shells but not in the fossil
345 faunas suggest that they likely present a seasonal character with a production period focussed in late summer as a response to environmental and trophic conditions. This is supported by previous studies in the Arctic where *T. quinqueloba* has been

found to dominate assemblages sampled in August (Carstens et al., 1997; Volkmann, 2000) but not in June/ early July (Pados and Spielhagen, 2014), and by sediment trap observations from the subpolar North Atlantic where *T. quinqueloba* reaches its maximum in autumn (Jonkers et al., 2010). The dominance of *N. pachyderma* in the fossil faunas collected at the BSO and its low but constant presence in the coloured shells of surficial sediment and plankton tow sampled in late summer 2014 suggests that this species may demonstrate a more sustainable behaviour with a regular production throughout the year. This hypothesis is supported by a recent study showing that abundances and distribution of the species *N. pachyderma* are not significantly perturbed by seasonal seawater temperature, productivity or salinity variations occurring in the Arctic (Greco et al., 2019). Thus it makes sense that *N. 
[revised manuscript text omitted]

---

## Referee Report (RR1)

Meilland et al. present an interesting study on the population dynamics of modern planktic foraminifera in the Western Barents Sea. As planktic foraminifera are a sensitive indicator of environmental changes in the Arctic, a region particularly affected by global temperature changes, the authors address a relevant scientific question within the scope of BG. The authors use a novel multiproxy approach combining the use of plankton net, core-top, molecular biology, environmental parameters and phytoplankton characterization. The manuscript has certainly been improved in response to the previous referees' comments. However, in my opinion, it still needs some minor revisions. I further suggest that the manuscript is read and corrected by an English native speaker or a professional author service due to numerous linguistic issues, only some of which I have listed below. In some places, they make following the manuscript difficult. I am looking forward for the authors' response and further discussion.

**General comments:**

Your vertical plankton hauls sampled only the uppermost 100 m, while for example, N. pachyderma can live as deep as 280 m (Greco et al., 2019). This might perhaps partly explain the discrepancy between living and fossil assemblages and should be discussed in the manuscript.

The authors interchangeably use terms such as "fossil assemblages", "core-top assemblages" or "assemblages in surficial sediments". It is not clear whether all these terms mean the same or not. Please be more consistent in using these terms.

Another issue is that some information given in Material and Methods are repeated in the Results (see specific comments below but please also check the entire manuscript for repetitions).

Specific/technical comments:

Keywords: I suggest adding "planktonic foraminifera" as a keyword

1. Introduction

The introduction is somewhat mixed-up. Paleoceanographic information is mixed with modern assemblage studies, habitat depth mixed with seasonal variability, foraminifera with other organisms... In some places, too much details is given (e.g., 104 tow hauls in Greco et al. 2019). Suddenly the Southern Indian Ocean pops out... Please consider rewriting to better structure this section.

40: Add a comma after "As a response"

45-46: Either "Such phenomenon (...) is" or "Such phenomena (...) are"

48-51: I suggest deleting the sentence "Eynaud (2011) noticed..." as irrelevant to the study of modern assemblages.

54: Change "observed PF by the means of plankton tows," to "analyzed PF collected with plankton tows". Please note that Pados and Spielhagen (2014) analyzed both forams living in Polar and Atlantic waters and used both plankton tows and core top samples.

73: Change "planktonic foraminifera" to PF (be consistent in using the abbreviations that you introduced)

79: Change "planktonic foraminifera from a same species" to "PF of the same species"

2. Oceanographic setting

84: Change "Spitsbergen Banken and shallow Bjørnøya; Storfjordrenna and Bjømøyrenna glacial troughs" to "Storfjordrenna and Bjørnøyrenna glacial troughs separated by shallow Spitsbergen Banken". Please verify if it's Spitsbergen Banken or Spitsbergenbanken.

87: Nothing is written about the currents carrying Arctic Water to the study area.

From 88 onwards: To my understanding, the Oceanographic section should only contain the state-of-the-art on the subject. If the authors performed some oceanographic measurements, please move the information to "Material and Methods" and "Results" sections.

98: "above described" => "described above"

3. Material and Methods

112 and elsewhere: You use either "Spitsbergen", "Spitzbergen" or "Spitzberg". Please unify. To my knowledge, "Spitsbergen" is the English spelling, while "Spitzbergen" is German and "Spitzberg" French.

118: Change "collection" to e.g., "sampling strategy"

124: "All living PF" – if they were preserved with ethanol, they were not living anymore. Change to, e.g., "All foraminiferal tests containing coloured cytoplasm ("living")...". Where the samples stained with Rose Bengal? If so, this should be mentioned. Otherwise, how were they coloured?

130: "separately numbered" => "counted separately"

137: "Individual" => "The individuals"

146: Referee#2 suggested using "core-top" instead of "subfossil". I think it also concerns the term "fossil".

147:  $INSU^1$  – please check if the journal accepts footnotes.

149: Delete "the more horizontal" – something is horizontal or not, it can't be more or less horizontal.

149-150: Change "The core-top sediment (0- 0.5 cm slice)" to "The uppermost 0.5 cm of the core"

152-156: Please rewrite the two sentences so that they are more related to each other.

4. Results

162: CTD, not CDT! "5 values" – what values? Please specify.

163-166: The paragraph gives absolutely no information about the results and most (if not all) the info were already given in the methods.

169: The highest PF concentrations were found at the edge of the NwCW range (station 7) so I would refrain from saying that the highest concentrations were found in NwCW.

173: It should be specified which species are considered polar and which subpolar by the authors.

188: "analyse" => "analysis" or "analyses"

194-195: The information in brackets was already given in the methods and is unnecessary here.

195: "successfully" – I assume you wouldn't mention them at all in the manuscript if they were unsuccessful.

204: Is it exactly equal (down to  $0.00000001 \ \mu m$ ) or close enough to saz that the siye distribution of the picked tests is szmmetric?

206: μm => μg

207:  $\mu g => \mu g / \mu m (or \, \mu g^* \mu m^{-1})$

209: "slightly (but not significantly)" – this is not very specific, please rephrase by, e.g., giving some numbers

212-215: This was already written in the methods.

217: Please add station numbers to Figure 7.

218: I would rather write that N. pachyderma was the most abundant species. Dominance suggests that it reached >50% which is the case only in one station.

5. Discussion

234-236: I don't understand the sentence. Please rephrase.

238: In my opinion, low abundances at the two ends of the transect do not suggest patchiness. Low abundances in the middle of the transect would suggest it.

274: Shouldn't it be station 7 instead of 8?

275: The location of the Polar Front should be marked in Figure 1.

6. Conclusions

363-365: Please specify that the percentages concern the living (plankton haul) population.

375: "is" => "in"

---

## Author Response (AR2)

Meilland et al. present an interesting study on the population dynamics of modern planktic foraminifera in the Western Barents Sea. As planktic foraminifera are a sensitive indicator of environmental changes in the Arctic, a region particularly affected by global temperature changes, the authors address a relevant scientific question within the scope of BG. The authors use a novel multiproxy approach combining the use of plankton net, core-top, molecular biology, environmental parameters and phytoplankton characterization. The manuscript has certainly been improved in response to the previous referees' comments. However, in my opinion, it still needs some minor revisions. I further suggest that the manuscript is read and corrected by an English native speaker or a professional author service due to numerous linguistic issues, only some of which I have listed below. In some places, they make following the manuscript difficult. I am looking forward for the authors' response and further discussion.

A : First of all we would like to thank the reviewer for her/his very helpful and constructive comments. The manuscript has been checked by two native English speaker and improved. We hope this new version of it will be satisfying.

General comments:

Your vertical plankton hauls sampled only the uppermost 100 m, while for example, N. pachyderma can live as deep as 280 m (Greco et al., 2019). This might perhaps partly explain the discrepancy between living and fossil assemblages and should be discussed in the manuscript.

A : we sampled down to 700 m depth for 7 of the 9 studied stations and found very low abundances below 100 m depth (<5 ind.m$^{-3}$) including for *N. pachyderma* abundances (< 1.25 ind.m$^{-3}$). We added this information to the manuscript and will make those counts available on Pangea as well.

The authors interchangeably use terms such as "fossil assemblages", "core-top assemblages" or "assemblages in surficial sediments". It is not clear whether all these terms mean the same or not. Please be more consistent in using these terms.

A : we went through the manuscript and change terms in order to be consistent.

Another issue is that some information given in Material and Methods are repeated in the Results (see specific comments below but please also check the entire manuscript for repetitions).

A : we removed these sections from the results

Specific/technical comments:

Keywords: I suggest adding „planktonic foraminifera" as a keyword

A : We added „planktonic foraminifera" in the keyword

1. Introduction

The introduction is somewhat mixed-up. Paleoceanographic information is mixed with modern assemblage studies, habitat depth mixed with seasonal variability, foraminifera with other

organisms... In some places, too much details is given (e.g., 104 tow hauls in Greco et al. 2019). Suddenly the Southern Indian Ocean pops out... Please consider rewriting to better structure this section.

A : we worked on the introduction in order to structure it better and keep what we consider important information.

40: Add a comma after "As a response"

A : comma added

45-46: Either "Such phenomenon (...) is" or "Such phenomena (...) are"

A : correction made

48-51: I suggest deleting the sentence "Eynaud (2011) noticed..." as irrelevant to the study of modern assemblages.

A : we deleted the sentence

54: Change "observed PF by the means of plankton tows," to "analyzed PF collected with plankton tows". Please note that Pados and Spielhagen (2014) analyzed both forams living in Polar and Atlantic waters and used both plankton tows and core top samples.

A : we replaced the sentence

73: Change "planktonic foraminifera" to PF (be consistent in using the abbreviations that you introduced)

A : correction made

79: Change "planktonic foraminifera from a same species" to "PF of the same species"

A : correction made

2. Oceanographic setting

84: Change "Spitsbergen Banken and shallow Bjørnøya; Storfjordrenna and Bjømøyrenna glacial troughs" to "Storfjordrenna and Bjørnøyrenna glacial troughs separated by shallow Spitsbergen Banken". Please verify if it's Spitsbergen Banken or Spitsbergenbanken.

A: we made the corrections L.91 and after verification in the literature, decided to use Spitsbergenbanken (correction also made L.102 for this term)

87: Nothing is written about the currents carrying Arctic Water to the study area.

A : Indeed we didn't mention Arctic Water in the paper as none of the discussed samples were collected in this water layer. Arctic Waters were only encountered twice during the cruise and mixed with NwAW (Giraudeau et al., 2016).

From 88 onwards: To my understanding, the Oceanographic section should only contain the state-of-the-art on the subject. If the authors performed some oceanographic measurements, please move the information to "Material and Methods" and "Results" sections.

A : we changed these sections of the manuscript accordingly

98: "above described" => "described above"

A : correction made

3. Material and Methods

112 and elsewhere: You use either "Spitsbergen", "Spitzbergen" or "Spitzberg". Please unify. To my knowledge, "Spitsbergen" is the English spelling, while "Spitzbergen" is German and "Spitzberg" French.

A : we thank the reviewer for this observation and unified using Spitsbergen

118: Change "collection" to e.g., "sampling strategy"

A : we amended following the reviewer's comment

124: "All living PF" – if they were preserved with ethanol, they were not living anymore. Change to, e.g., "All foraminiferal tests containing coloured cytoplasm ("living")...". Where the samples stained with Rose Bengal? If so, this should be mentioned. Otherwise, how were they coloured?

A : the net samples were not stained and individuals that was living at the moment of the sampling (the one we call "living") were distinguished by the natural coloration of their cytoplasm. We therefore choose not keep this sentence intact.

130: "separately numbered" => "counted separately"

A : correction made

137: "Individual" => "The individuals"

A : correction made

146: Referee#2 suggested using "core-top" instead of "subfossil". I think it also concerns the term "fossil".

A : we disagree and would prefer to keep fossil here.

147: INSU[1] – please check if the journal accepts footnotes.

A : we removed the footnote and place detailed the accronym INSU directly in the text.

149: Delete "the more horizontal" – something is horizontal or not, it can't be more or less horizontal.

A : we replaced horizontal by "even"

149-150: Change "The core-top sediment (0- 0.5 cm slice)" to "The uppermost 0.5 cm of the core"

A : we replaced following the Reviewer's suggestion

152-156: Please rewrite the two sentences so that they are more related to each other.

A : this section has been modified

4. Results

162: CTD, not CDT! "5 values" – what values? Please specify.

A : correction and specification made

163-166: The paragraph gives absolutely no information about the results and most (if not all) the info were already given in the methods.

A : the paragraph has been slightly rewritten but kept as we think it provides guidance to the reader.

169: The highest PF concentrations were found at the edge of the NwCW range (station 7) so I would refrain from saying that the highest concentrations were found in NwCW.

A : we changed the sentence to precise highest concentrations were found at the edge of the NwCW.

173: It should be specified which species are considered polar and which subpolar by the authors.

A : We used the SCOR WG138 taxonomy for this study where only *N. pachyderma* is described as a polar species. As the used taxonomy is mentioned in the Material and Methods, we would rather not comment on it in this section.

188: "analyse" => "analysis" or "analyses"

A : corrected

194-195: The information in brackets was already given in the methods and is unnecessary here.

A : we removed the breckets

195: "successfully" – I assume you wouldn't mention them at all in the manuscript if they were unsuccessful.

A : we removed "successfully"

204: Is it exactly equal (down to 0.00000001 μm) or close enough to saz that the siye distribution of the picked tests is szmmetric?

A : we added precisions to the text

206: μm => μg

A : correction made

207: μg => μg/μm (or μg*μm$^{-1}$)

A : correction made

209: "slightly (but not significantly)" – this is not very specific, please rephrase by, e.g., giving some numbers

A : we mentioned the Figure 6 once again here.

212-215: This was already written in the methods.

A : we removed it from this section

217: Please add station numbers to Figure 7.

A : Figure 7 amended

218: I would rather write that N. pachyderma was the most abundant species. Dominance suggests that it reached >50% which is the case only in one station.

A : we agree with the Reviewer and corrected

5. Discussion

234-236: I don't understand the sentence. Please rephrase.

A : we changed the sentence

238: In my opinion, low abundances at the two ends of the transect do not suggest patchiness. Low abundances in the middle of the transect would suggest it.

A : we disagree with the reviewer

274: Shouldn't it be station 7 instead of 8?

A : we made the correction

275: The location of the Polar Front should be marked in Figure 1.

A : we prefer to keep Figure 1 as simple as possible and would rather not add the location of the Polar front. We cited Oziel et al., 2017 here, presenting very accurate data for it. Moreover, our transect don't cross the polar front.

6. Conclusions

363-365: Please specify that the percentages concern the living (plankton haul) population.

A : we provided this additional information

375: "is" => "in"

A : we did the correction

[revised manuscript text omitted]